# Vps11 and Vps18 of Vps-C membrane traffic complexes are E3 ubiquitin ligases and fine-tune signalling

Gregory Segala [1], Marcela A. Bennesch[1], Nastaran Mohammadi Ghahhari[1], Deo Prakash Pandey[1,3], Pablo C. Echeverria [1], François Karch [2], Robert K. Maeda[2] & Didier Picard [1]

In response to extracellular signals, many signalling proteins associated with the plasma membrane are sorted into endosomes. This involves endosomal fusion, which depends on the complexes HOPS and CORVET. Whether and how their subunits themselves modulate signal transduction is unknown. We show that Vps11 and Vps18 (Vps11/18), two common subunits of the HOPS/CORVET complexes, are E3 ubiquitin ligases. Upon overexpression of Vps11/Vps18, we find perturbations of ubiquitination in signal transduction pathways. We specifically demonstrate that Vps11/18 regulate several signalling factors and pathways, including Wnt, estrogen receptor α (ERα), and NFκB. For ERα, we demonstrate that the Vps11/18-mediated ubiquitination of the scaffold protein PELP1 impairs the activation of ERα by c-Src. Thus, proteins involved in membrane traffic, in addition to performing their well-described role in endosomal fusion, fine-tune signalling in several different ways, including through ubiquitination.

[1] Département de Biologie Cellulaire, Université de Genève, Sciences III, 30 quai Ernest-Ansermet, 1211 Genève, Switzerland. [2] Département de Génétique et Évolution, Université de Genève, Sciences III, 30 quai Ernest-Ansermet, 1211 Genève, Switzerland. [3]Present address: Department of Molecular Microbiology, Oslo University Hospital, 0372 Oslo, Norway. Correspondence and requests for materials should be addressed to D.P. (email: didier.picard@unige.ch)

Cells are permanently sensing the signals from the extracellular medium to adapt their identity and their actions to the local environment. The sensors of these signals are membrane or cytoplasmic receptors, which can specifically bind signaling molecules, such as growth factors or steroid hormones. Upon activation, transient signaling complexes are assembled and signal transduction is accomplished through consecutive steps of stimulatory and inhibitory post translational modifications of effectors, involving mainly phosphorylation and ubiquitination[1–3]. One of the final outcomes is the regulation of the activity of specific transcriptional complexes. A striking example is ERα, a nuclear receptor that responds to both estrogen and other signal transduction pathways to regulate the expression of target genes as transcription factor[4]. A plethora of enzymes that modify signal transducers have been described[2]. However, most of the post translational modifications identified by proteome-wide analyses have not yet been linked to specific enzymes[5]. This suggests that many enzymes regulating signal transduction remain to be discovered.

Most signal transduction pathways start at the plasma membrane through receptor activation and the formation of membrane-tethered signaling complexes[1]. A subset of ERα molecules associate with the plasma membrane for extranuclear non-genomic signaling[6–8]; estrogen binding triggers the rapid formation of an active signaling complex involving the scaffolding proteins BCAR1 and PELP1[6,9,10], connecting membrane-associated ERα and kinases such as the tyrosine kinase c-Src, leading to their activation.

Recently, it was shown that the endosomal machinery creates clusters of membrane signaling complexes whose activities are controlled by endosomal fusion[11,12]. Endosomal fusion depends on the class C core vacuole/endosome tethering (CORVET) and homotypic fusion and protein sorting (HOPS) that are hexameric complexes conserved from yeast to humans; they share the class C core consisting of the subunits Vps11, Vps16, Vps18, and Vps33[13,14]. Studies on CORVET, HOPS, and their individual subunits have mainly focused on the description of their roles in endosomal fusion[15–24] and autophagy[25–27]. Intriguingly, within the class C core, the two subunits Vps11 and Vps18 possess a C-terminal RING domain[20,28,29]. The RING domain defines the most widespread family of E3 ubiquitin ligases, a set of enzymes which catalyze the coupling of ubiquitin to substrate proteins[30]. Indeed, it has been reported that Vps18 can act as an E3 ubiquitin ligase[31,32], but the general roles of both Vps11 and Vps18 (hereafter referred to as Vps11/18) and their enzymatic activities have not been described.

In this study, we characterize the ubiquitinomes of Vps11/18 and demonstrate that these two factors are involved in the regulation of signal transduction by protein ubiquitination. To provide a more in-depth understanding of this regulatory mechanism, we characterize the regulation of the transcriptional activity of ERα by Vps11/18 in more detail. We show that Vps11/18 inhibit the formation of the ERα membrane complex with c-Src by preventing its interaction with PELP1 through ubiquitination. This impairs a feedforward stimulation between membrane and nuclear ERα molecules.

## Results

**Vps11/18 are E3 ubiquitin ligases involved in signalling.** RING domains like those found in Vps11/18 (Fig. 1a) are described for both E3 ubiquitin ligases and E3 SUMO ligases[33]. To determine whether Vps11/18 possess either one of these activities, we overexpressed Vps11/18 in HEK293T cells and examined whether bulk ubiquitination (Fig. 1b) or sumoylation were increased (Supplementary Fig. 1a, b). The overexpression of Vps11/18 strongly increased the amount of ubiquitinated proteins, whereas it did not affect the levels of sumoylated proteins. As expected, we did not observe any change of bulk ubiquitination upon overexpression of Vps16 or Vps33A. To gain further insights into the functions of Vps11/18, we designed a proteomic strategy to characterize the Vps11/18 ubiquitinomes (Fig. 1c). We labeled cells of the control condition with light amino acids (L), whereas cells overexpressing Vps11/18 were labeled with heavy amino acids (H)[34]. The cleavage of ubiquitinated sites by trypsin leaves a diGly moiety from the ubiquitin that can be recognized by a specific antibody to enrich for originally ubiquitinated peptides[35]. The results show that 70 and 119 sites were significantly more ubiquitinated when Vps11 and Vps18 were overexpressed, respectively (Fig. 1d, e). Unexpectedly, Vps11 and Vps18 also downregulated the ubiquitination of 117 and 141 sites, respectively. Overall, Vps11/18 overexpression resulted mostly in single-site ubiquitination (Fig. 1f), and Vps11 and Vps18 modified the ubiquitination of 106 and 148 proteins, respectively (Fig. 1f and Supplementary Data 1). Of the 56 proteins, whose ubiquitination is affected by both Vps11 and Vps18 (Fig. 1g), we identified three E3 ubiquitin ligases that are more ubiquitinated (indicated by arrows). One of them is UBE2O, a hybrid ubiquitin-conjugating enzyme/E3 ubiquitin ligase, which has been reported to target ribosomal proteins for degradation[36,37]. Interestingly, most of the downregulated ubiquitination sites in our datasets belong to ribosomal proteins, suggesting that Vps11/18 E3 ligase activities indirectly decreased the ubiquitination of many proteins by targeting other E3 ubiquitin ligases. We can formally not exclude that some of the ubiquitinome changes, both increases and decreases, are due to indirect and non-catalytic effects of Vps11/Vps18; however, based on experiments with RING mutants of Vps11/Vps18 presented below, we assume that many or most effects may be attributable to their E3 ligase activities.

To get a sense of the global functions of the Vps11/18 E3 ligase activities, we combined a gene set enrichment analysis (GSEA) of the ubiquitinated sites with a gene ontology (GO) study. We generated a representation of the interactome that clusters the proteins, whose ubiquitination is affected by either Vps11 or Vps18, by GO terms (Fig. 1h). The clusters protein synthesis and protein degradation are oppositely regulated by Vps11/18. This suggests that Vps11/18 may control proteostasis by ubiquitination. For the cluster signal transduction, the upregulation of ubiquitination suggests that Vps11/18 directly ubiquitinate signal transduction substrates. Moreover, a detailed analysis of the most significant GO terms highlighted several signaling pathways, such as the Wnt signaling pathway (Supplementary Fig. 1c–h). We decided to focus on the involvement of the E3 ubiquitin ligase activities of Vps11/18 in the regulation of signal transduction.

**Several signaling pathways are regulated by Vps11/18.** To assess the functional roles of Vps11/18 in signal transduction, we investigated the impact of altering their expression levels on a panel of signal transduction pathways. The knock-down of individual Vps-C core subunits may affect the assembly or stability of the entire HOPS/CORVET complexes. In the human cell line HEK293T, we therefore checked the stability of HOPS/CORVET complexes with native gels by probing immunoblots for Vps33A when Vps11/18 are silenced or overexpressed (Fig. 2a). A band slightly above 545 kDa corresponds to the theoretical sizes of the HOPS/CORVET complexes of 582 and 640 kDa, respectively. The knock-down of Vps11/18 with two different shRNAs (Fig. 2a and Supplementary Fig. 2a, b) decreased the amount of complexes while the overexpression of Vps11/18 did not affect them. In contrast, the simultaneous overexpression of all Vps-C core components strongly increased the amount of complexes.

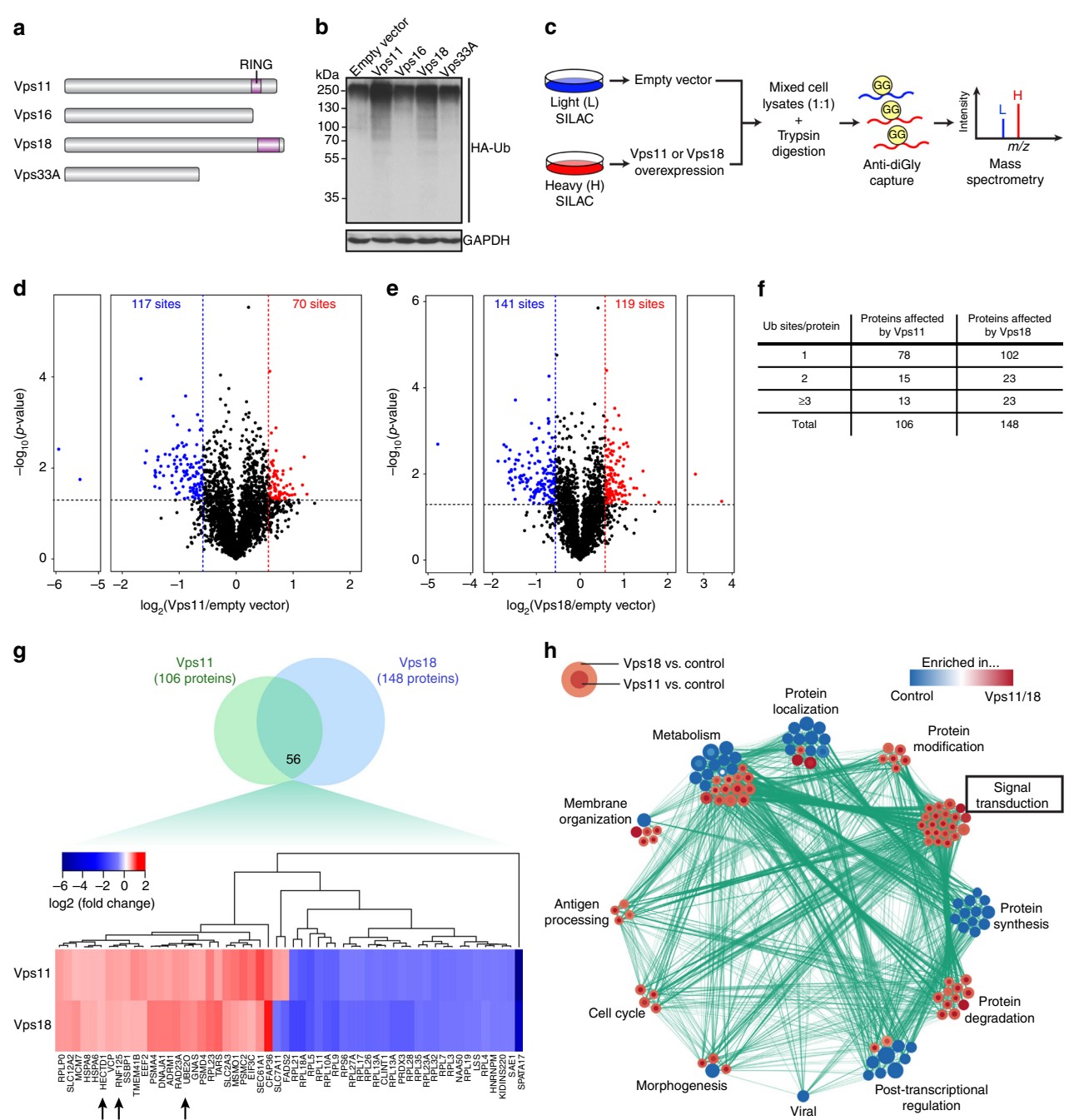

**Fig. 1** Vps11/18 carry E3 ubiquitin ligase activities involved in signal transduction. **a** Scheme of the class C Vps proteins that constitute the core of the HOPS and CORVET complexes. The RING domain is highlighted in purple. **b** In vivo ubiquitination assay with HEK293T cells overexpressing the indicated Vps proteins along with HA-tagged ubiquitin. Ubiquitinated proteins were immunoblotted (IB) with an HA antibody; GAPDH is the internal loading control. **c** Experimental strategy to quantify the ubiquitinome upon overexpression of Vps11/18 in HEK293T cells using stable isotope labeling with amino acids in cell culture (SILAC) and mass spectrometry; see text for more details. **d**, **e** Volcano plots showing the differentially ubiquitinated sites (log$_2$ ratios) between cells transfected with the empty expression vector and cells overexpressing either Vps11 (**d**) or Vps18 (**e**) and their statistical significance ($-\log_{10}(p\text{-value})$ on the y-axis); values above the black dotted line are significantly regulated; significantly downregulated sites (blue dots) are at the top left of the blue threshold line while significantly upregulated sites (red dots) are at the top right of the red threshold line. **f** Table of the number of protein substrates of Vps11/18 with one or several differentially regulated ubiquitination sites. **g** Venn-diagram of the differentially ubiquitinated proteins in Vps11- or Vps18-overexpressing cells compared to the empty vector condition (at the top) with the zoomed in shared proteins ordered by hierarchical clustering (at the bottom); arrows point to E3 ubiquitin ligases that are more ubiquitinated upon Vps11 or Vps18 overexpression. **h** GO-based clustering of the interactome of the proteins associated with a significant change of the ubiquitinome upon Vps11/18 overexpression. Each node represents a GO term with a color code showing the relative changes of the corresponding ubiquitinome upon Vps11 (inner circle) or Vps18 (outer circle) overexpression (blue or red for enriched in control cells over cells overexpressing Vps11/18 or the other way around, respectively). The number of proteins with a particular GO term is represented by the size of its corresponding node. Source data are provided as a Source Data file

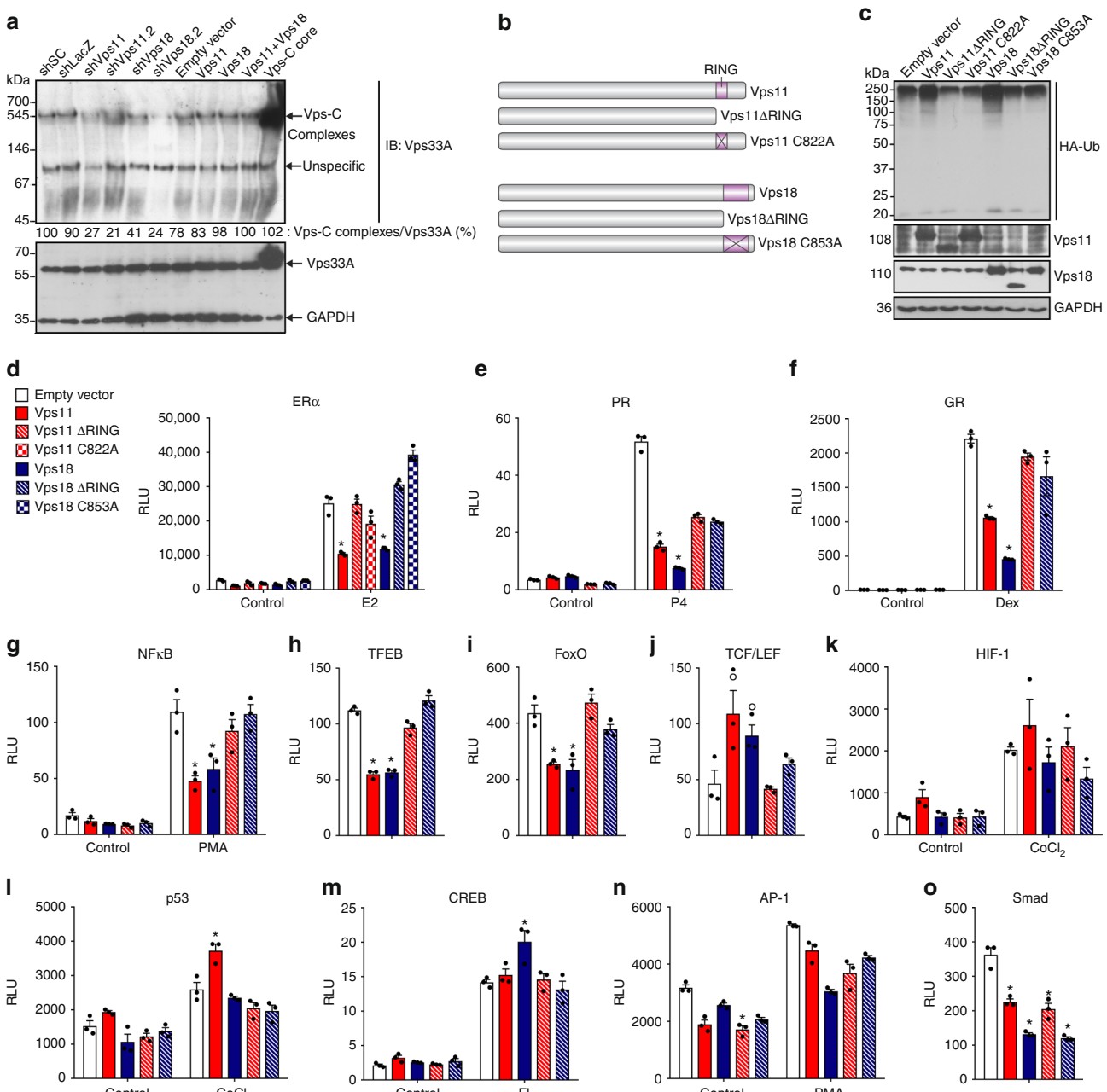

**Fig. 2** The E3 ubiquitin ligase activities of Vps11/18 influence signal-regulated transcription. **a** Immunoblot of a native protein gel of Vps-C complexes of HEK293T cells with either a shRNA-mediated knock-down or overexpression of Vps11/18. The simultaneous overexpression of all Vps-C core components was used as a positive control. A scrambled shRNA (shSC) and a shRNA targeting the bacterial β-galactosidase (shLacZ) were used as negative controls for the knock-downs. The immunoblot was probed with a Vps33A antibody. In parallel, an immunoblot was performed on the same samples separated under denaturing conditions to determine the total amount of Vps33A. The indicated Vps-C complexes/Vps33A ratios were measured with the software ImageJ and are representatives of two immunoblots. **b** Scheme of the RING domain mutants of Vps11/18. A cross inside the RING domain indicates defective point mutants. **c** In vivo ubiquitination assay of the RING mutants as in Fig. 1b. **d–o** Luciferase reporter assays for the indicated transcription factors, performed in HEK293T cells overexpressing either Vps11/18 or their corresponding RING domain mutants. For ERα, PR, GR, and CREB signaling pathways, the transcriptional activities were induced by a treatment with E2, progesterone (P4), dexamethasone (Dex) or a cocktail of forskolin and isobutylmethylxanthine (FI) to increase intracellular cAMP levels, respectively; NFκB and AP-1 were induced by phorbol myristate acetate (PMA), HIF-1 and p53 with cobalt (II) chloride ($CoCl_2$). RLU, relative light units. Bar graphs show the averages ± s.e.m. with $n \geq 3$ biologically independent experiments. Asterisks indicate significant differences with the corresponding negative controls with $p$-values < 0.05; the dots above some of the bars in **j** indicate differences with $p$-values of 0.05–0.06. Statistical significance was determined with unpaired and two-sided Student's $t$-tests. Source data are provided as a Source Data file

We conclude that overexpression of individual components is a better experimental strategy, easily feasible in mammalian cells, to study specifically Vps11/18 without disturbing the levels of the HOPS/CORVET complexes. This allowed us to use truncation and point mutants of the RING domains of Vps11/18 (Fig. 2b) to question specifically the role of their E3 ubiquitin ligase activity. We confirmed the loss of E3 ubiquitin ligase activities for all of the mutants of Vps11/18 (Fig. 2c), and ascertained that their overexpression did not affect the total amount of assembled HOPS/CORVET complexes (Supplementary Fig. 2c).

We next tested the impact of the overexpression of wild-type and mutant Vps11/18 on transcriptional activities depending on signaling pathways identified in the GSEA-GO analysis (Supplementary Fig. 1c–h); to this end, we performed transfection experiments using luciferase reporter constructs for a panel of transcription factors (Fig. 2d–o). Vps11/18 repressed the activity of the steroids receptors ERα, progesterone receptor (PR) and glucocorticoid receptor (GR), of NFκB, TFEB, and FoxO in a RING domain-dependent fashion (Fig. 2d–i). Interestingly, the Wnt signaling pathway reported by the activity of the transcription factor TCF/LEF is stimulated by Vps11/18 overexpression, and the E3 ubiquitin ligase activities are also essential for this regulation (Fig. 2j). The activities of hypoxia inducible factor 1α (HIF-1), p53, cAMP-responsive element binding protein (CREB), and AP-1 are affected by either Vps11 or Vps18 and this depends on their respective RING domains (Fig. 2k–n). In contrast, the transforming growth factor β (TGFβ) signaling pathway (Smad) is repressed by Vps11/18 independently of the RING domain and its E3 ubiquitin ligase activities (Fig. 2o).

Having seen that several signaling pathways involved in development are regulated by Vps11/18, we assessed whether downregulation of Vps-C components affects development in the fruit fly *Drosophila melanogaster*. Note that *Drosophila* Vps18 (dVps18, also known as Deep Orange[38]) had already been linked to Wnt signaling in flies[39]. Our results show that the downregulation of any of the Vps-C components in either the posterior compartment or the dorsal compartment impaired the proper development of the posterior or the dorsal part of wings, respectively (Supplementary Fig. 3a). We found that expression levels of Vps-C components increase in third-instar larvae (Supplementary Fig. 3b), a specific stage during fly development that is associated with a strong activity of the ecdysone signaling pathway. Indeed, the downregulation of Vps-C components strongly decreased the expression of ecdysone receptor targets (Supplementary Fig. 3c–f). Hence, this argues that in flies the HOPS/CORVET complexes rather than an independent activity of Vps11/18 are necessary for ecdysone signaling.

For further mechanistic studies, we decided to focus on the unexpected regulation of certain pathways by Vps11/18 in an E3 ubiquitin ligase-dependent way. We chose ERα as a model transcription factor because it is well established as a target of many signal transduction pathways[4,7]. We found that repression of ERα is a specific activity of Vps11/18 as the overexpression of the other Vps-C components Vps16 or Vps33A or their combination did not affect ERα activity (Fig. 3a). Similarly, the overexpression of Vps8 or Vps41, two other subunits containing RING-like domains, specific of CORVET and HOPS, respectively, had no effect (Supplementary Fig. 4a). The combination of Vps11 and Vps18 overexpression repressed ERα similarly showing that the regulation of ERα activity by Vps11 and Vps18 is largely redundant (Fig. 3a). We further confirmed with the knock-down of Vps11/18, using two different shRNAs each, that Vps11/18 are repressors of ERα (Fig. 3a and Supplementary Fig. 4b) and GR (Supplementary Fig. 4c) activities independently of their roles in HOPS/CORVET complexes, as the knock-down of Vps16 and Vps33A did not affect ERα and GR (Fig. 3b and Supplementary

Fig. 4b, c). For ERα, these results were confirmed by assessing the effects of Vps11/18 levels on a few representative endogenous ERα target genes in ERα-positive breast cancer cells. Similarly to what we had seen with exogenous ERα in HEK293T cells (see Fig. 3a and Supplementary Fig. 4d), the knock-down and overexpression of Vps11/18 in MDA-MB-134 breast cancer cells increased and decreased expression of endogenous ERα target genes, respectively (Fig. 3c–e and Supplementary Fig. 4e, f). Note that repression of endogenous ERα target genes by Vps11/18 could be demonstrated with MCF-7 breast cancer cells as well, indicating that the phenomenon is independent of a specific cell line.

**ERα is regulated by Vps11/18 through a specific pathway**. We hypothesized that the regulation of ERα by Vps11/18 may involve intracellular membrane traffic because ubiquitination had been described to control processes from endocytosis to late endosomes[3,40]. Directed perturbation of endocytosis with a dominant-negative mutant of dynamin II (Supplementary Fig. 5a) or with a knock-down of caveolin-1 did not counteract the repression of ERα by Vps11/18 (Fig. 4a, b and Supplementary Fig. 5b). Overexpression of Vps11/18 did not perturb the uptake of transferrin, further supporting the conclusion that endocytosis is not involved in the repression of ERα activity by Vps11/18 (Fig. 4c). As CORVET and HOPS complexes are effectors of the GTPases Rab5 and Rab7[41], we knocked them down as well, but this did not affect the regulation of ERα by Vps11/18 either (Fig. 4b and Supplementary Fig. 5b). Furthermore, the overexpression of Vps11/18 or their corresponding E3 ligase mutants did not affect trafficking from early endosomes to lysosomes as judged by co-staining for the markers EEA1 and LAMP1, respectively, with the endocytosed transferrin (Fig. 4d and Supplementary Fig. 5c). Moreover, the silencing of HRS and TSG101, two ubiquitin-binding proteins critically involved in the sorting of ubiquitinated proteins into intraluminal vesicles from the late endosome[40], had no effect on Vps11/18-mediated repression of ERα (Fig. 4e and Supplementary Fig. 5b). We also tested the involvement of the Golgi apparatus, the lysosome and the proteasome with the inhibitors brefeldin A, chloroquine and MG132, respectively, (Supplementary Fig. 5d, e); again, these drugs did not prevent the repression of ERα by Vps11/18. Furthermore, we ruled out an involvement of autophagy by blocking it with wortmannin or 3-methyladenine (3-MA), or by stimulating it by starvation or with rapamycin (Fig. 4f, g and Supplementary Fig. 5f). Likewise, the knock-down of SQSTM1 to inhibit selective autophagy[42] had no effect on the repression of ERα by Vps11/18 (Fig. 4h and Supplementary Fig. 5b). We next hypothesized that Vps11/18 might affect the EGF receptor (EGFR), mitogen-activated protein kinase (MAPK) or the protein kinase A (PKA) signal transduction pathways. We inhibited EGFR with AG1478, but even though it reduced the activity of ERα by half, Vps11/18 were still able to repress ERα (Fig. 4i). The phosphorylation of PKA substrates was not modified by Vps11/18 overexpression whereas the phosphorylation of the MAPKs ERK1/2 was increased by Vps11 but not by Vps18, indicating that they do not mediate the effects of Vps11/18 on ERα (Fig. 4j, k). The overexpression of the transcriptional coactivators SRC1 and CARM1 or of the transcriptional corepressor NCoR1[43] did not affect the repression of ERα by Vps11/18 showing that the regulation may not directly depend on an effect on nuclear coregulators (Fig. 4l and Supplementary Fig. 5g). Thus, Vps11/18 do not globally affect signal transduction, but rather target individual signaling pathways in a specific fashion.

**Vps11/18 directly ubiquitinate PELP1 to control ERα activity**. Having excluded a whole panel of mechanisms regarding the

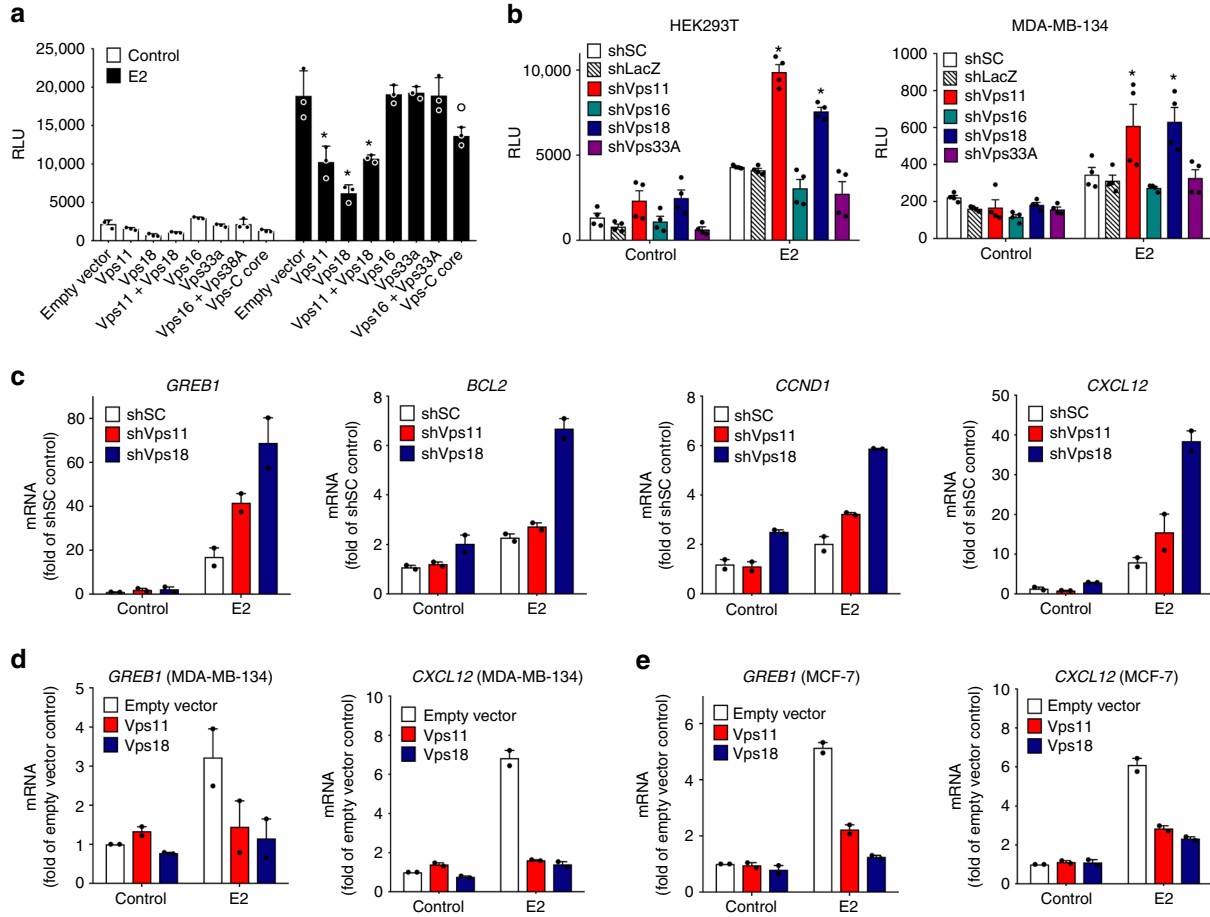

**Fig. 3** ERα transcriptional activity is specifically repressed by Vps11/18. **a** ERα reporter assay with HEK293T cells overexpressing different combinations of Vps-C core components and treated or not with E2 (mean ± s.e.m. with $n \geq 3$ biologically independent experiments). **b** ERα reporter assays with HEK293T or MDA-MB-134 breast cancer cells infected with lentiviral shRNA constructs for the knock-down of Vps-C core components and treated or not with E2 (mean ± s.e.m. with $n \geq 3$ biologically independent experiments); see Supplementary Fig. 4b for the results with a second set of shRNA constructs. **c–e** Quantitative RT-PCR analysis of the mRNA levels of ERα target genes in MDA-MB-134 (**c**, **d**) or in MCF-7 cells (**e**) (mean ± s.e.m. with $n = 2$ biological independent experiments). Asterisks indicate significant differences with the corresponding negative controls with $p$-values < 0.05; the dot above the bar in **a** indicates a difference with a $p$-values of 0.05–0.06. Statistical significance was determined with unpaired and two-sided Student's $t$-tests. Source data are provided as a Source Data file

regulation of ERα by Vps11/18, we generated an in silico inter-actome of ERα, Vps11 and Vps18 to glean new ideas (Fig. 5a). We found BCAR1 as a common interactor of ERα and Vps11. BCAR1, ERα, PELP1, and c-Src form a complex at the plasma membrane depending on the palmitoylation of ERα on Cys447 tethering ERα to the plasma membrane[9,44]. Binding of 17β-estradiol (E2) to membrane ERα triggers the activation of c-Src[9,44] that can in turn stimulate the transcriptional activity of ERα itself by phosphorylation[45]. When we used a palmitoylation mutant of ERα that impairs membrane ERα signaling, we observed that the knock-down of Vps11/18 cannot stimulate its activity (Fig. 5b). The co-expression of the ligand binding domain of ERα, which is sufficient to recapitulate membrane ERα signaling[46], restored the effect of the knock-down of Vps11/18 on transcriptional activity of the full-length palmitoylation mutant of ERα. This confirms that membrane ERα signaling is required for the regulation of the transcriptional activity of ERα by Vps11/18. Remarkably, we found by co-immunoprecipitation that Vps11/18 interact with several proteins of the membrane-associated ERα complex (Fig. 5c). Of these proteins, and including ERα, only PELP1 was downregulated at the protein level by prolonged expression of Vps11/18 in HEK293T and MDA-MB-134 cells (Fig. 5d and Supplementary Fig. 6a), suggesting that PELP1 is

specifically targeted by Vps11/18. It is important to note that except for ERα, which was ectopically expressed in HEK293T cells, all proteins of the membrane ERα complex are endogenously present in both cell lines. A ubiquitination assay with PELP1 and BCAR1 showed that PELP1 is ubiquitinated by overexpressed Vps11/18, but BCAR1 is not (Fig. 5e). As PELP1 is known to be SUMOylated[47], we used a mutant of the SUMOy-lation site of PELP1, but this did not affect the regulation of ERα by Vps11/18 (Supplementary Fig. 6b). We also tested if SUMOylation is more generally involved by using a dominant-negative mutant of UBC9, the universalSUMO conjugase[33], but its overexpression did not affect the regulation of ERα by Vps11/18 (Supplementary Fig. 6c). This indicates that only ubiquitina-tion of PELP1 is involved. Intriguingly, we failed to find a ubi-quitinated site for PELP1 in our proteomic data. To explain this apparent paradox, we hypothesized that the ubiquitination site of PELP1 might be present in a short tryptic peptide of six amino acids or less, which cannot be efficiently analyzed by mass spectrometry (MS). We predicted the theoretical tryptic peptides associated with each lysine of PELP1 (Supplementary Fig. 6d), and for the peptides of less than ten amino acids, we mutated the corresponding lysines to arginine. Overexpression of the single mutant K496R of PELP1 not only impaired the repression of ERα

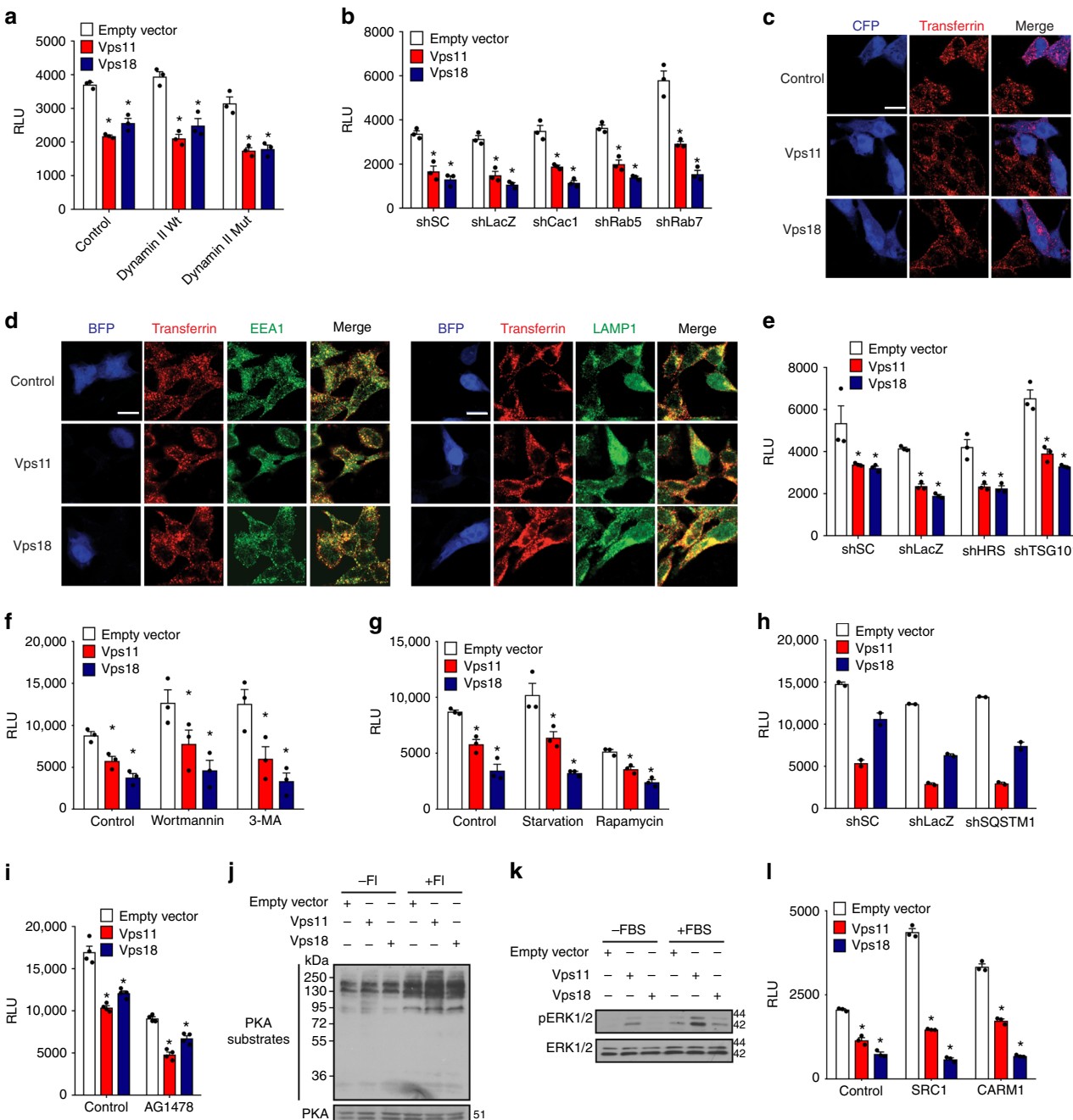

**Fig. 4** Regulation of ERα activity by Vps11/18 is independent of intracellular trafficking pathways. **a** ERα reporter gene assays with HEK293T cells overexpressing Vps11/18 in combination with wild-type (Wt) or K44A mutant (Mut) dynamin II (mean ± s.e.m. with $n = 3$ biologically independent experiments). **b** Assays as in **a** but with knock-downs with the indicated shRNA constructs (mean ± s.e.m. with $n = 3$ biologically independent experiments). **c** Transferrin uptake assays in HEK293T cells overexpressing Vps11/18 along with cyan fluorescent protein (CFP). Scale bar indicates 20 μm. **d** Transferrin uptake assays with immunostaining of EEA1 or LAMP1 in HEK293T cells overexpressing Vps11/18 and blue fluorescent protein (BFP). Scale bar indicates 20 μm. **e** Assays as in **a** with knock-downs of HRS or TSG101 (mean ± s.e.m. with $n = 3$ biologically independent experiments). **f–i** Assays as in **a** under the following conditions: treatments to inhibit (Wortmannin, 3-methyladenine (3-MA)) or to stimulate (serum starvation, rapamycin) autophagy (**f** and **g**, respectively); knock-down of SQSTM1 (**h**); treatment with the EGFR inhibitor AG1478 (**i**) (mean ± s.e.m. with $n \geq 3$ biologically independent experiments). **j, k** Immunoblots displaying PKA substrates and protein levels with or without FI (**j**) or phospho-ERK1/2 and ERK1/2 protein levels with or without FBS (**k**) in HEK293T cells overexpressing Vps11/18; FI, cocktail of forskolin and isobutylmethylxanthine to increase intracellular cAMP levels; FBS, fetal bovine serum. **l** Assays as in **a** with overexpression of the transcriptional coactivators SRC1 or CARM1 (mean ± s.e.m. with $n = 3$ biologically independent experiments). Asterisks indicate significant differences with the corresponding negative controls with $p$-values < 0.05. Statistical significance was determined with unpaired and two-sided Student's $t$-tests. Source data are provided as a Source Data file

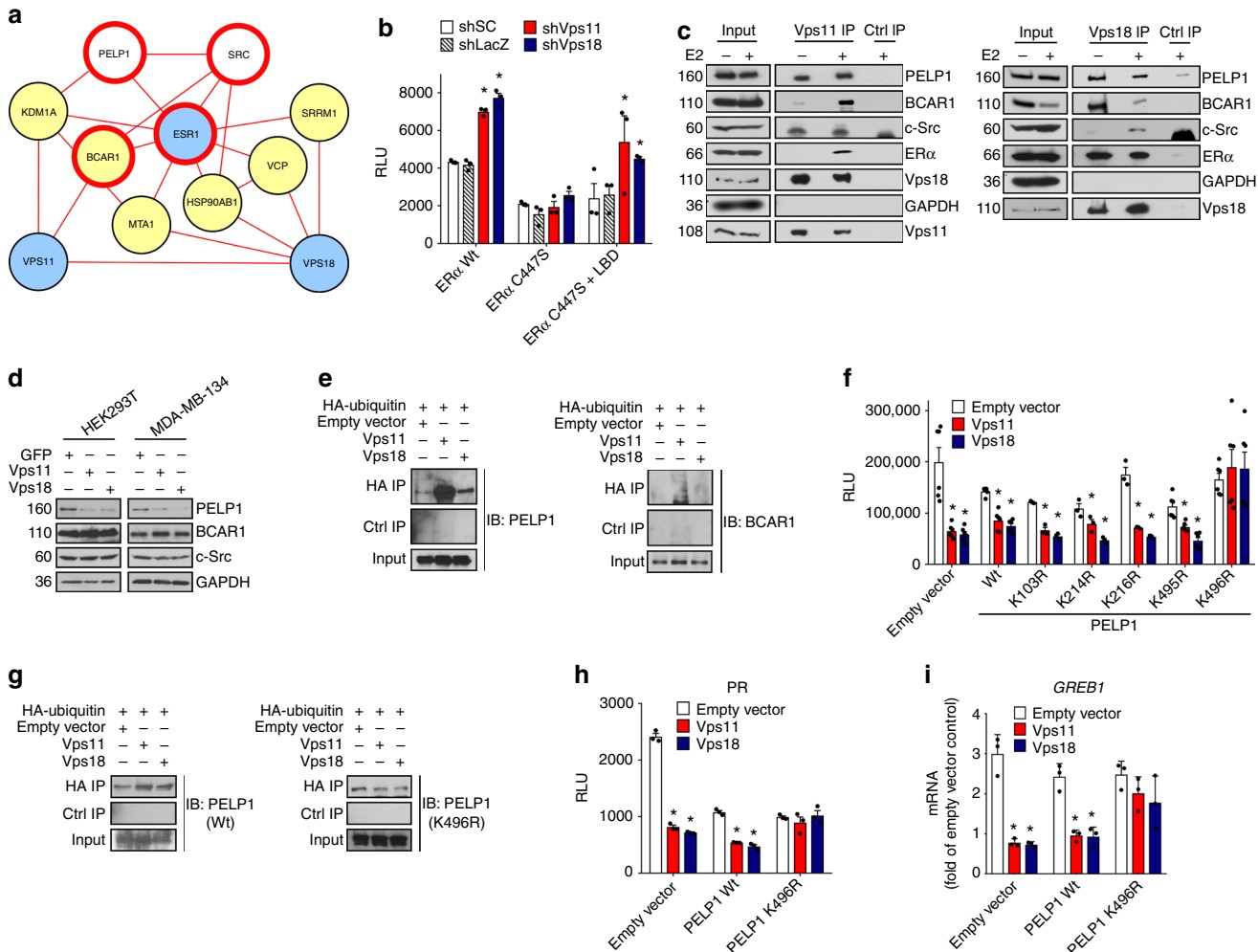

**Fig. 5** Vps11/18 prevent membrane-associated ERα signaling by ubiquitinating PELP1. **a** Combined interactome of ERα (ESR1) and Vps11/18 (blue nodes) generated with Cytoscape using the plugin PpiMapBuilder (https://goo.gl/GusMZG). The common interactors are in yellow. Proteins in white nodes do not directly interact with Vps11/18 and nodes with a bold red line represent the proteins that are associated with the membrane ERα pathway. **b** ERα reporter assays with HEK293T cells infected with the indicated shRNA constructs and overexpressing either wild-type ERα (Wt), an ERα palmitoylation mutant (ERα C447S) or the combination of full-length ERα C447S with the ligand binding domain of ERα (LBD) by itself. Cells were treated with E2 (mean ± s.e.m. with n = 3 biologically independent experiments). **c** Immunoblot of an immunoprecipitation experiment with extracts from MDA-MB-134 cells. A control immunoprecipitation was performed in parallel with a control IgG (Ctrl IP). The blots were probed for the proteins shown on the right. **d** Immunoblots of extracts from HEK293T and MDA-MB-134 cells overexpressing Vps11/18 for an extended period of 6 days. **e** Immunoblots of immunoprecipitated ubiquitinated proteins. Extracts from HEK293T cells overexpressing Vps11/18 along with HA-tagged ubiquitin were immunoprecipitated with an HA antibody and probed either for endogenous PELP1 (left) or BCAR1 (right). A control immunoprecipitation was performed in parallel with a control IgG (Ctrl IP). **f** ERα reporter assays with HEK293T cells overexpressing Vps11/18 along with wild-type (Wt) PELP1 or the indicated point mutants of PELP1. **g** In vivo ubiquitination assays as in **e** with exogenously expressed wild-type and mutant PELP1 as indicated. **h** Luciferase reporter assays for PR in HEK293T cells overexpressing Vps11/18 along with wild-type or mutant PELP1 (mean ± s.e.m. with n = 3 biologically independent experiments). P4, progesterone. **i** Quantitative RT-PCR analysis of the mRNA levels of the ERα target gene *GREB1* in MDA-MB-134 cells overexpressing Vps11/18 along with the indicated proteins (mean ± s.e.m. with n = 3 biologically independent experiments). Asterisks indicate significant differences with the corresponding negative controls with p-values < 0.05. Statistical significance was determined with unpaired and two-sided Student's t-tests. Source data are provided as a Source Data file

by Vps11/18 (Fig. 5f), it also failed to be ubiquitinated by Vps11/18 (Fig. 5g). When we determined the effects of this mutant on other signaling pathways, we found that the regulation of PR by Vps11/18 depends on K496 of PELP1 (Fig. 5h), whereas that of NFκB does not (Supplementary Fig. 6e). These results point out that the mechanism of regulation of signal transduction by Vps11/18 is specific for each signaling pathway. We confirmed with MDA-MB-134 cells that K496 of PELP1 is required for the regulation of endogenous ERα by Vps11/18 as the overexpression of the PELP1 mutant K496R impairs the repression of the endogenous ERα target gene *GREB1* by Vps11/18 (Fig. 5i).

**Vps11/18 inhibit ERα phosphorylation by c-Src.** Within the membrane ERα complex, PELP1 acts as a scaffold bringing together c-Src and membrane ERα, and thereby activating the kinase activity of c-Src[6,9] in response to E2. Whereas the overexpression of PELP1 did not affect the regulation by Vps11/18, overexpression of a mutant of PELP1 that is unable to interact with c-Src[48] mimicked the repression of ERα by Vps11/18 and prevented further repression by them (Fig. 6a). Importantly, this result also points out that variations of the PELP1 levels, at least within certain limits, do not account for the effects of Vps11/18 on ERα; instead, this regulation seems

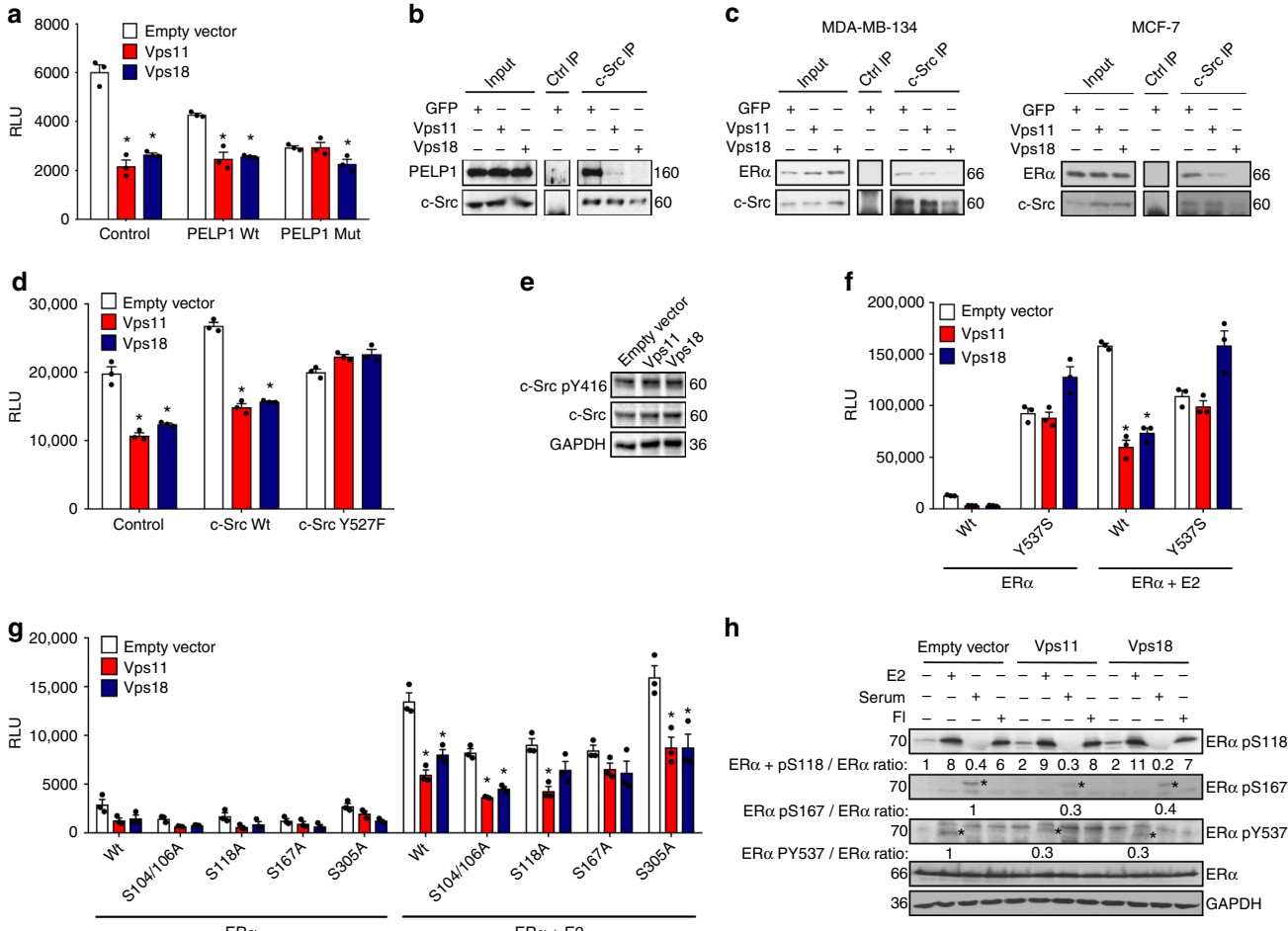

**Fig. 6** Vps11/18 regulate the phosphorylation of ERα by c-Src. **a** Luciferase assay with an ERα-dependent luciferase construct in HEK293T cells overexpressing Vps11/18 along with PELP1 Wt or a mutant of PELP1[48] unable to interact with c-Src (PELP1 Mut) (mean ± s.e.m. with $n \geq 3$ biologically independent experiments). **b**, **c** PELP1 (**b**) or ERα immunoblot (**c**) of the immunoprecipitations of c-Src with extracts of MDA-MB-134 (**b**, **c**) or MCF-7 cells (**c**) transiently overexpressing either GFP or Vps11/18 for only 3 days (unlike the prolonged expression in the experiments of Fig. 5d) and treated with E2. **d** ERα reporter gene assays with HEK293T cells overexpressing Vps11/18 along with either wild-type c-Src (Wt) or the constitutively active c-Src mutant Y527F (mean ± s.e.m. with $n = 3$ biologically independent experiments). **e** Immunoblots of phospho-c-Src (Y416) and total c-Src with extracts of HEK293T cells overexpressing Vps11/18. **f**, **g** Asssays as in **d** with the indicated tyrosine (**f**) or serine (**g**) phosphorylation mutants of ERα (mean ± s.e.m. with $n = 3$ biologically independent experiments). **h** Immunoblots with antibodies specific for ERα phosphorylation sites with extracts of MDA-MB-134 cells overexpressing Vps11/18; before harvesting, cells were treated with E2, serum or FI for 10 min where indicated; asterisks point out the specific ERα bands; the indicated ratios of phosphorylated ERα/total ERα were determined with the software ImageJ on representative immunoblots. Asterisks indicate significant differences with the corresponding negative controls with p-values < 0.05. Statistical significance was determined with unpaired and two-sided Student's t-tests. Source data are provided as a Source Data file

to depend on the capacity of PELP1 to interact with c-Src. Indeed, we showed with MDA-MB-134 cells that Vps11/18 overexpression inhibited the interaction between PELP1 and c-Src (Fig. 6b). We further predicted that Vps11/18 overexpression should compromise PELP1 as a scaffold and not only affect its own interaction with c-Src. By immunoprecipitating c-Src, we were able to confirm that the interaction of c-Src and ERα is reduced by Vps11/18 overexpression both in MDA-MB-134 and MCF-7 cells (Fig. 6c). This suggests that the ubiquitination of PELP1 by Vps11/18, in these short-term experiments even before it leads to its degradation (compare Fig. 5d and Supplementary Fig. 6a with Fig. 6b), prevents the activation of c-Src by PELP1 in the membrane ERα complex. In line with this hypothesis, we found that upon over-expression of a constitutive mutant of c-Src ERα activity became insensitive to the repression by Vps11/18 (Fig. 6d), even though the global activity of c-Src was not affected by Vps11/18 (Fig. 6e). This indicates that it is the specific activity

of c-Src within the membrane ERα complex that is regulated by Vps11/18 through PELP1. E2 treatment induces the phosphorylation of ERα on Y537 by c-Src, and this phosphorylation is essential for ERα activity[49]. Therefore, we tested the phosphorylation mutant Y537S of ERα (Fig. 6f) and found, as expected, that it could not be repressed by Vps11/18; other phosphorylation mutants of ERα were still sensitive to the regulation by Vps11/18 (Fig. 6g). With immunoblotting experiments using phosphorylation-specific antibodies, we found that Vps11/18 repressed the phosphorylation of S167 and Y537 (Fig. 6h), showing that other kinases in addition to c-Src are affected, but that of the main phosphorylation sites of ERα only Y537 is essential for the repression by Vps11/18. Taken together, our results lead us to propose that Vps11/18 ubiquitinate PELP1 on K496 preventing c-Src from interacting with and being activated by membrane-associated ERα. Without activated c-Src, ERα is not phosphorylated on Y537 and its transcriptional activity is decreased.

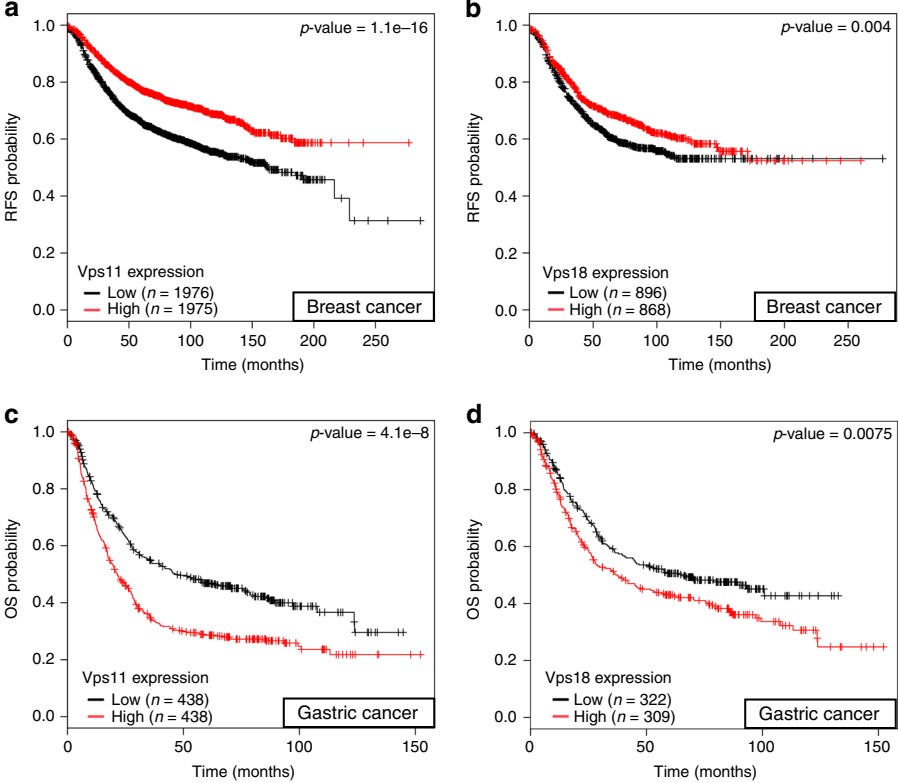

**Fig. 7** Vps11/18 define a cancer type-specific clinical outcome. **a–d** Kaplan–Meier survival analyses of different cancer types as a function of gene expression levels of Vps11 (**a**, **c**) or Vps18 (**b**, **d**). **a**, **b** Relapse-free survival (RFS) analyses of patients with breast tumors. **c**, **d** Overall survival (OS) analyses of patients with gastric tumors

**Vps11/18 levels predict cancer patient survival**. Considering the fact that Vps11/18 are regulators of ERα, they could be prognostic markers for ERα-positive breast cancer patients. Indeed, we found that breast cancer patients with high expression levels of Vps11/18 have a higher probability of relapse-free survival (Fig. 7a, b), For gastric cancer, which is ERα-independent, the clinical value of high expression of Vps11/18 is exactly opposite (Fig. 7c, d). The involvement of Vps11/18 in cancer initiation and progression may be cancer type-specific and linked to their specific effects on the underlying signaling pathways.

## Discussion

We have discovered a role for the E3 ubiquitin ligase activities of Vps11/18 in signal transduction that is independent of their role as components of the HOPS/CORVET complexes. This was unexpected despite the fact that membrane traffic and fusion of endosomes were known to be linked to signal transduction[12]. HOPS/CORVET might be generally involved in signal processing by the endosomal machinery, without any specificity for particular signaling pathways. Some signaling pathways, whose regulation strongly depends on the endosomal machinery, might be more affected by the loss of function of HOPS/CORVET; indeed, these membrane trafficking complexes might be required for EGFR signaling in vertebrates[11], and we have shown here that they are essential during wing development and for ecdysone signaling in *Drosophila*. For other signaling pathways, cells may exploit the enzymatic activities of the core subunits Vps11/18 for specific regulatory purposes.

This is strikingly illustrated by the regulation of c-Src by Vps11/18: while they do not change the global activity of c-Src, the subset of c-Src molecules activated in a PELP1-dependent way is affected, leading to reduced phosphorylation of ERα on Y537

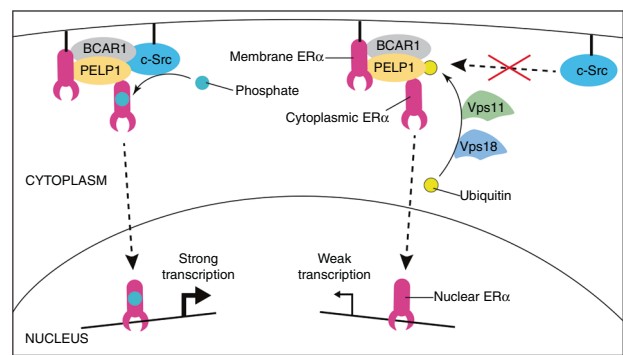

**Fig. 8** Proposed mechanism for the regulation of ERα activity by Vps11/18. Note that the purple symbol for ERα represents both the monomer and the dimer of ERα

and reduced ERα transcriptional activity. The membrane complex associated with a small subset of ERα molecules, which are mostly nuclear, provides a relay system by which estrogen signalling integrates non-genomic effects at the membrane with genomic effects in the nucleus (Fig. 8). The dashed arrow in Fig. 8 indicates the fact that the mode of communication between membrane and nuclear ERα activities remains unclear. The feedforward stimulation may be mediated by the rapid non-genomic activation of kinases, which in turn stimulate the nuclear activities of ERα molecules already in the nucleus; alternatively, ERα molecules themselves might cycle between ERα membrane complexes or transient encounters with the activated membrane-associated kinase c-Src, while overall nuclear accumulation persists[6–8]. Either way, Vps11/Vps18 could fine-tune this feedforward loop by targeting PELP1. The key event is the ubiquitination

of PELP1 and the ensuing disruption of the complex with c-Src, long before PELP1 levels drop, possibly due to proteasome-mediated degradation. Whether a similar mechanism is involved in the regulation of other transcription factors such as GR and PR (Supplementary Fig. 3), will have to be investigated in more molecular detail. For GR and PR, non-genomic signalling is well established and PELP1 has at least been linked to GR[10,50–52].

We could demonstrate that the regulation of ERα by Vps11/18 does not depend on the endosomal machinery. Instead, the ubiquitination of PELP1 by Vps11/18 is sufficient to prevent its interaction with c-Src and the subsequent phosphorylation of ERα by c-Src. It is very likely that other substrates of Vps11/18 are regulated through their binding to the ubiquitin receptors Hrs and/or Tsg101, resulting in their sequestration into intraluminal vesicles inside endosomes. Since this is a general mechanism for the endosomal regulation of proteins by ubiquitination[40], it is important to characterize the targets of E3 ligases such as Vps11/18 more comprehensively.

As components of the HOPS/CORVET complexes, Vps11 and Vps18 form the stalk of the two-headed structures and interact with complex-specific accessory subunits that allow the tethering of membranes of different compartments[41]. There are some interesting functional analogies between the accessory subunits that are on the same side of HOPS/CORVET. On the Vps11 end, the TGFβ receptor-associated protein 1 (TGFBRAP1, also known as Vps3) for CORVET and Vps39 for HOPS are known to be involved in the regulation of the TGFβ signaling pathway by interacting directly with Smad proteins[53,54]. On the Vps18 end, Vps8 for CORVET and Vps41 for HOPS are also RING finger proteins[41]. Although Vps8 and Vps41, in contrast to Vps11/18, do not regulate the activity of ERα, we cannot exclude that Vps8 and Vps41 regulate other signaling pathways. Intriguingly, the topological orientation of Vps11 and Vps18 in CORVET and HOPS complexes is such that their respective RING domains are both engaged in interactions with accessory subunits[28]. It remains to be elucidated whether they nevertheless somehow exert their activities as E3 ligases as part of the complexes. Alternatively, the RING domains might only play an architectural role within the complexes, but allow E3 ligase of unincorporated Vps11/Vps18 moonlighting away from the HOPS/CORVET complexes in regulating signal transduction. The importance of these RING domains, be it as structural components or as E3 ligases, is clearly illustrated by several clinical cases showing that mutations in the RING domain of Vps11 are associated with a variety of developmental and neurological disorders[55–57]. It is interesting to consider that despite a large proportion of proteins whose ubiquitination is similarly affected by Vps11 and Vps18 overexpression, more than half of the affected proteins are not shared. The specific substrate spectrum of Vps11/Vps18 could be shaped by differences in their enzymatic preferences or cofactors, or, if they were able to function within the HOPS/CORVET complexes, by a functional polarity imposed by these host complexes. This could provide the cell with an additional opportunity to regulate signal transduction in time and space.

Our work expands previous knowledge of the endosome as an essential organelle in signal transduction[3,12,40,58]. We have provided evidence for another level of regulation at the crossroads between endosomal fusion and post translational modification of signal transducers that further intertwines these cellular events. This may prompt further studies to explore how the endosomal machinery and factors potentially cycling on and off the corresponding membrane traffic complexes integrate and regulate the inputs from multiple signaling pathways.

## Methods

### Antibodies and reagents.
The rabbit polyclonal antisera against ERα (HC-20, sc-543, discontinued) and p-ERα (Ser118, sc-12915-R, discontinued), the mouse monoclonal antibodies against ERK1/2 (C-9, sc-514302), p-ERK (E-4, sc-7383), Vps11 (S-38, sc-100893) and SUMO-2/3/4 (C-3, sc-393144), and the goat polyclonal anti-Vps16 (C-17, sc-86939, discontinued) were from Santa Cruz Biotechnologies (all diluted 1/200 for immunoblots); the rabbit polyclonal antisera against PELP1/MNAR (A300-180A) and BCAR/p130Cas (A301-667A) were from Bethyl Laboratories (all diluted 1/500 for immunoblots); the mouse monoclonal anti-GAPDH (6C5, ab8245) was from Abcam (diluted 1/30,000 for immunoblots); the mouse monoclonal anti-HA.11 (16B12, MMS-101P) was from Biolegend (for immunoprecipitations, 2 μg of antibody was used for 2 mg of proteins); the rabbit polyclonal antisera against Vps33A (PA545268), p-ERα (Ser167, PA537570) and p-ERα (Tyr537, PA537571), the mouse monoclonal antibody against Vps18 (4E9, MA522391) were from Thermo Fisher Scientific (all diluted 1/500 for immunoblots; for immunoprecipitations, 2 μg of anti-Vps18 antibody were used for 2 mg of proteins); the rabbit polyclonal antiserum against PKA substrates (P-(S/T), 9621), the rabbit monoclonal antibody against phospho-Src (Tyr416) (D49G4, 6943) (both diluted 1/500 for immunoblots); the mouse monoclonal anti-c-Src (GD11, 05–184) was from Millipore (diluted 1/500 for immunoblots); the rabbit polyclonal antiserum against SUMO-1 was from Alexis Biochemicals (BML-PW0505A, diluted 1/500 for immunoblots); the rabbit polyclonal antibody against human EEA1 (ALX-210–239) was from Enzo (diluted 1/100 for immunofluorescence), and the mouse monoclonal antibody against human LAMP1 (H4A3) was from BD PharMingen (diluted 1/100 for immunofluorescence).17β-estradiol (used at 100 nM), dexamethasone (used at 100 nM), progesterone (used at 100 nM), phorbol myristate acetate (PMA) (used at 1 μg/ml), cobalt (II) chloride (used at 100 μM), wortmannin (used at 1 μM), 3-methyladenine (used at 5 mM), rapamycin (used at 1 μM), forskolin (used at 10 μM), isobutylmethylxanthine (used at 100 μM), chloroquine (used at 50 μM), brefeldin A (used at 5 μg/ml) and AG1478 (used at 10 μM) were from Sigma-Aldrich; MG132 (used at 5 μM) was from Enzo Life Sciences. Transferrin conjugated to AlexaFluor 680 was from Thermofisher.

### Cell culture.
Human embryonic kidney (HEK293T, ATCC CRL-3216), human breast carcinoma cells MCF-7 (ATCC HTB-22), and MDA-MB-134[59] (a gift from Wilbert Zwart, Netherlands Cancer Institute, Amsterdam) were maintained in Dulbecco's Modified Eagle's Medium (DMEM) supplemented with 10% fetal bovine serum and 1% penicillin/streptomycin. For transfection and induction experiments, cells were cultured for at least 72 h before induction in DMEM without phenol red supplemented with 5% charcoal-stripped fetal bovine serum, 2 mM L-glutamine and 1% penicillin/streptomycin (white medium). All cell lines were regularly tested to be mycoplasma negative.

### Plasmids.
The following expression vectors were used: p3xFlag-CMV-10 (Sigma), pcDNA3.1(+) (Invitrogen), pSG5[60] and pCMV5[61]. The luciferase reporters were the following: EREtkLuc (also called XETL)[62] for ERα, GREtkLuc (also called XG46TL)[62] for GR, the pGL2-based construct PRE-TATA-Luc (a gift from D. McDonnell, Duke University) for PR, FHRE-Luc for FOXO3a (a gift from Michael Greenberg, Harvard Medical School; Addgene plasmid #1789)[63], M50 Super 8x TOPFlash for TCF/LEF (a gift from Randall Moon, University of Washington; Addgene plasmid #12456)[64], PG13-Luc for p53 (a gift from Bert Vogelstein, Johns Hopkins University; Addgene plasmid #16442)[65], SBE4-Luc for Smad (a gift from Bert Vogelstein; Addgene plasmid #16495)[66], HRE-Luc for HIF-1α (a gift from Navdeep Chandel, Northwestern University; Addgene plasmid #26731)[67], 4XCLEAR-Luciferase for TFEB/TFE3 (a gift from Albert La Spada, UC San Diego; Addgene plasmid #66800)[68], pUC-(TRE)5TL[69] for AP-1, pNFkB-Luc (Stratagene) for NFkB, pCRE-Luc (Stratagene) for CREB and the Renilla luciferase transfection control pRL-CMV (Promega). Additional plasmids: pECFP-N3 and pEBFP-N1, pRK5-HA-Ubiquitin (a gift from Ted Dawson, Johns Hopkins University; Addgene plasmid #17608)[70], pcDNA3.2/V5-DEST-Vps8 (a gift from Judith Klumperman, UMC Utrecht), pCMV-SPORT6-Vps11 (Transomics), pCMV-SPORT6-Vps16 (Transomics), pGFP-C3-mVps18 (a gift from R. Piper, University of Iowa)[71], pCA-Nflag-Vps18 (a gift from Y. Kawaoka, University of Tokyo)[29], pcDNA3.2/V5-DEST-Vps41 (a gift from Judith Klumperman), pGFP-C3-Dynamin II and p3xFlag-Dynamin II K44A (gifts from A. Roux, University of Geneva), pcDNA3-UBC9 and pcDNA3-UBC9 C93S[72], pEBG-BCAR1 (a gift from Raymond Birge, Cancer Institute of New Jersey; Addgene plasmid #15001)[73], pCMV-ERα C447S (a gift from R. Miksicek, Michigan State University)[74], pSG5-SRC1 (a gift from M. Parker, Imperial College London), pSG5-HA-CARM1 (a gift from M. Stallcup, University of Southern California)[75], pEBG-PELP1 and pEBG-PELP1-Mut-SRC (gifts from R. Vadlamudi, University of Texas Health Science Center at San Antonio)[48], pSGT-c-Src (a gift from G. Superti-Furga, Research Center for Molecular Medicine of the Austrian Academy of Sciences), pcDNA3-c-Src Y527F (a gift from G. Gallick, University of Texas MD Anderson Cancer Center)[76], pCMV-ERα Y537S (a gift from B. Katzenellenbogen, University of Illinois)[77], HEG0 for expression of wild-type ERα[78], and pSG5-hPR[79] for expression of PR. For expression of the hormone binding domain of ERα fused to GFP, we inserted appropriate coding sequences into expression vector pNEF, which contains the strong EF-1α promoter region, yielding plasmid pNEF/F.ER. The expression vectors pCMV-hGR for GR and pCMV-Vps33A were obtained by inserting the coding

regions for human GR and human Vps33A, respectively, into expression vector pCMV5. Expression vectors for Vps11 and Vps18 were generated by inserting the human Vps11 and the mouse Vps18 ORFs into plasmid pHAGE-CMV-fullEF1a-IRES-ZsGreen (plasmid ID 233 from the DNA Resource Core at the Harvard Medical School, Boston). Deletion mutants of the RING domains of Vps11 (Vps11ΔRING) and Vps18 (Vps18ΔRING) were generated by isolating by PCR the regions of the ORFs corresponding to amino acids 1–821 for Vps11 and 1–852 for Vps18 and inserting them into expression vector pcDNA3.1(+). Point mutations of Vps11, Vps18, PELP1, and ERα were produced by the QuickChange method. The shRNA constructs were generated with vector pLKO.1 (Open Biosystems) according to the details given in Supplementary Table 1. Lentiviruses were generated with plasmids pMD2G and psPAX2 (a gift from Didier Trono, Ecole Polytechnique Fédérale de Lausanne (EPFL)).

**Lentivirus production and cell transduction**. HEK293T cells were seeded to a density of 1.5 millions per 100 mm-dish in standard medium 24 h before poly-ethylenimine (PEI) transfection with plasmids pMD2G, psPAX2, and the shRNA-encoding pLKO plasmids. Sixteen hours later, the medium was changed to white medium and lentivirus-containing supernatants were collected every 8–12 h during 36 h. Cells were infected by the lentivirus-containing supernatants during 72 h. After infection, cells were collected for experiments. To avoid phenotype variations upon long-term shRNA-mediated knock-downs, newly infected sets of cells were used each time.

**Stable isotope labeling by amino acids in cell culture**. Stable isotope labeling by amino acids in cell culture (SILAC) was performed as follows: Isotope-labeled amino acids ($^{13}C_6^{15}N_2$-L-lysine, $^{13}C_6^{15}N_4$-L-arginine, >99%, Cambridge Isotope Laboratories (CIL), Andover, MA) were included in the heavy-SILAC medium at 100 mg/l, whereas proline was supplied at 180 mg/l (a ninefold excess over its standard concentration in RPMI medium) in all media. Heavy or light-SILAC labeling was achieved by culturing the cells for a minimum of 2 weeks to allow for at least five cell divisions. Before the start of the experiments, tests were carried out to verify that heavy labeling was greater than 98% and Arg to Pro conversion was lower than 3%. Heavy-SILAC labeled cells were transfected with Vps11- or Vps18-coding plasmids and light-SILAC-labeled cells were transfected with the empty vector pCMV5 and used as a control. Four hours before harvesting the cells, the proteasome inhibitor MG132 and the lysosomal inhibitor chloroquine were added to each cell dish at a final concentration of 5 μM and 10 μM, respectively.

**Protein sample preparation for mass spectrometry**. The procedures we used for protein sample preparation, MS, and data analysis are very similar to those previously reported by Quadroni and colleagues[80]. The following gives the detailed protocol used for this particular study. One-hundred twenty million cells per condition were used with three technical replicates, which yielded approximately 11 mg of total protein per replicate. Cells were harvested, washed twice with phosphate-buffered saline (PBS) and lysed by pulse sonication in 8 M urea, 50 mM Tris pH 7.5 and Phos-stop™ phosphatase inhibitors (Roche). After clarification by centrifugation at 15,000 × g, heavy and light extracts were quantitated and mixed at a molar ratio of 1:1 to obtain 22 mg of total protein before proceeding with digestion essentially as specified in the protocol for the PTMscan system (Cell Signaling Technology, Danvers, MA, USA). Briefly, proteins were reduced by incubation with 4.5 mM DTT for 1 h at room temperature followed by alkylation of cysteines with 30 mM chloroacetamide in the dark at room temperature. After dilution of the solution to 2 M urea, digestion was performed overnight at 37 °C by adding 1:55 (400 μg) TPCK-treated bovine trypsin (Sigma-Aldrich). A second aliquot of 60 μg trypsin and 30 μg Lys-C (Promega) were added to reach a final protein:enzyme ratio of 1:45 and the digestion was continued for 6 h at 37 °C. Completeness of digestion was verified by sodium dodecyl sulfate polyacrylamide gel electrophoresis (SDS-PAGE). After acidification, digests were desalted on Sep-Pak C18 cartridges, peptides were eluted with 6.0 ml of 40% acetonitrile and lyophilized. Enrichment of GlyGly-modified peptides was carried out with the Ubiquitin remnant motif PTMscan kit (Cell Signaling Technology) according to the instructions provided by the manufacturer. After elution and lyophilization, peptides were resuspended in 1.4 ml 50 mM MOPS pH 7.2, 10 mM Na$_2$HPO$_4$, 50 mM NaCl, centrifuged and the supernatant incubated for 30 min with 80 μl of bead-bound antibody. The resin was washed and bound peptides were eluted with 2 × 55 μl of 0.15% trifluoroacetic acid. The eluate was desalted with a C$_{18}$ StageTip (Thermo Fisher Scientific). Dried peptides were resuspended in 0.1% formic acid, 2% (v/v) acetonitrile for injection.

**Mass spectrometry (MS)**. Samples were analyzed on an Orbitrap Fusion trihybrid mass spectrometer (Thermo Fisher Scientific) interfaced via a nanospray source to a Dionex RSLC 3000 nanoHPLC system (Dionex). Peptides were separated on a custom packed nanocolumn (75 μm ID × 40 cm, 1.8 μm particles, Reprosil Pur, Dr. Maisch) with a gradient from 5 to 55% acetonitrile in 0.1% formic acid in 120 min. Full MS survey scans were performed at 120,000 resolution. All survey scans were internally calibrated using the 445.1200 background ion mass. Every sample was analyzed three times with three different peptide fragmentation methods. A first general higher energy collision dissociation (HCD) method

targeted peptide charge states 2+ −5+ with a normalized collision energy of 32%. As most ubiquitinated peptides are charged 3+ or more, a second method was used, which selected only charge states 3+ −5+ . A third injection was done using a method that performed EThCD fragmentation, also on charge states 3+ −5+ , with activation energies at 25% (HCD) and 34% (ETD). All data were pooled per each sample. In data-dependent acquisition controlled by Xcalibur 2.1 software (Thermo Fisher Scientific), a maximum number of precursor ions was selected within a maximum cycle time of 3.0 s. All tandem MS spectra were measured in the Orbitrap at 15,000 resolution. Dynamic exclusion of precursors was for 60 s. The mass spectrometric proteomic data have been deposited to the ProteomeXchange Consortium (http://proteomecentral.proteomexchange.org) via the PRIDE[81] partner repository with identifier PXD009178.

**MS data analysis**. Data was analyzed and quantified using MaxQuant version 1.5.3.30, which uses the Andromeda search engine[82]. The human subset of the release 2015_12 (December 2015) of the UniProtKB database was used, together with a collection of sequences of common contaminants. Mass tolerances were of 4.5 ppm (after recalibration) for the precursor and 20 ppm for tandem mass spectra. Carbamidomethylation of cysteine was specified as a fixed modification, while oxidation of methionine and protein N-terminal acetylation were specified as variable modifications in addition to Lys Gly–Gly modification. Cleavage specificity was trypsin (cleavage after K, R) with two missed cleavages. Peptide and protein identifications were filtered at 1% false discovery rate (FDR) established by MaxQuant against a reversed sequence database. Sets of protein sequences, which could not be discriminated based on identified peptides, were listed together as protein groups. Details of peak quantitation and protein ratio computation and normalization by MaxQuant are described elsewhere[83]. GlyGly modification sites were filtered by applying an Andromeda score cutoff of 40. The FDR for GlyGly sites with the parameters used was 1.07% against reverse sequences and no further filtering was applied. As the enrichment of modified peptides was only partial, it was possible to determine with MaxQuant SILAC total ratios for a certain number of proteins (2145 with a minimum of two peptides) present in the samples with unmodified peptides. All subsequent filtering steps, statistics and annotation enrichment analyses were performed using the Perseus software[84]. A description of the data processing steps performed can be found in the ProteomeXchange repository, together with complete raw MaxQuant output tables.

Normalized *H/L* ratios were filtered to keep only GlyGly sites with number values in all three replicates. These values were then plotted against the $-\log_{10} p$-values for each site to generate the volcano plots. Significantly changed *H/L* ratios were aggregated to obtain the average *H/L* ratio for every proteins. Lists of Vps11 and Vps18 protein substrates were intersected to generate the Venn diagram. Average *H/L* ratios of common proteins in both Vps11 and Vps18 datasets were represented in the heatmap. For the gene set enrichment analysis (GSEA)[85], normalized *H/L* ratios were aggregated for each replicates to generate a table with the average *H/L* ratio for every proteins. This table was converted to the gct format and used with the GSEA 3.0 software along with the gene ontology gene sets, all from the Broad Institute, to perform the GSEA. GSEA tables were then analyzed with Enrichment Map[86] from Cytoscape to generate the GO interactome.

**Fly strains and handling**. *Drosophila melanogaster* flies were raised at 25 °C on a standard yeast-cornmeal-agar medium. All crosses were performed using standard *Drosophila* genetic techniques. For the knock-down experiments, the following transgenic lines were used: dVps11 RNAi [GD24731], dVps11 RNAi [KK107420], dVps16A RNAi [GD23769], dVps16A RNAi [330158], dVps18 RNAi [GD33734], dVps18 RNAi [KK107053], dVps33A RNAi [GD4548] and dVps33A RNAi [KK110756]. These lines were obtained from the Vienna Drosophila Resource Center (VDRC, Vienna, Austria). The Gal4 driver strains used were obtained from the Bloomington Stock Center (Bloomington, IN, USA) and are as follows: y[1] w[*]; P{w[+mC] = Act5C-GAL4}25FO1/CyO, y[+] (RRID:BDSC_4414), y[1] w[*]; P{w[+mW.hs] = en2.4-GAL4}e16E (RRID:BDSC_30564), and w[1118] P{w [+mW.hs] = GawB}Bx[MS1096] (RRID:BDSC_8860). RNA expression analysis were performed on larvae collected at the third-instar stage and then directed lysed for RNA extraction.

**RNA extraction**. Cells seeded in six-well plates or larvae were lysed with the guanidium-acid-phenol method by adding TRI reagent (4 M guanidium thiocyanate, 25 mM sodium citrate, 0.5% N-lauroylsarcosine, 0.1 M 2-mercaptoethanol, pH 7) directly onto the cells. Lysates were transferred to tubes. In all, 2 M Na-acetate pH-4, aquaphenol and then chloroform:isoamyl alcohol (49:1) were added to the lysates and vigorously mixed by vortexing. Organic and aqueous phases were separated by centrifugation at 10,000 × g for 5 min and the top phases were collected. RNA was precipitated by the addition of absolute isopropanol and a centrifugation at 16,000 × g for 20 min at 4 °C. RNA pellets were washed with 80% ethanol and centrifuged at 16,000 × g for 5 min at 4 °C. The pellets were dried at room temperature and resuspended in nuclease-free water. RNA concentrations, and the 260/280 and 260/230 ratios were measured with a Nanodrop.

**Reverse-transcription and quantitative PCR**. RNA extracts were digested with RNase-free DNase (Promega) according to the manufacturer's instructions. 400 ng

RNA were reverse-transcribed to cDNA with random primers (Promega), GoScript buffer (Promega) and reverse-transcriptase (Promega) according to the manufacturer's instructions. cDNAs were mixed with the GoTaq master mix (Promega) and with specific primer pairs (Supplementary Table 2) for real-time qPCR with a Biorad CFX96 thermocycler according to the manufacturer's instructions. RNA levels were normalized with *GAPDH* as the internal standard.

**Protein extraction and co-immunoprecipitation**. Cells were washed once and harvested with PBS, pelleted, and lysed with ice-cold lysis buffer (10 mM Tris-HCl pH 7.5, 50 mM NaCl, 1 mM EDTA, 1 mM DTT, 10% glycerol, 10 mM Na-molybdate, and protease inhibitor cocktail (Roche)). Cell lysates were sonicated during 15 cycles of 15 s at high power using a Bioruptor sonicator (Diagenode). Cell debris were discarded by centrifugation and protein concentrations were measured with the Bradford assay. For immunoprecipitation, 2 mg of proteins were incubated overnight at 4 °C on a rotating wheel with a specific antibody or a control antibody of the same species (control IgG). 20 μl of protein G-dynabeads (Life Technologies), equilibrated with the lysis buffer, were then added and incubated for 3 h at 4 °C. Dynabeads were harvested with a magnetic stand and washed three times with 0.1% Triton X-100 in lysis buffer followed by a wash with the lysis buffer only. Proteins were eluted with a reducing buffer (sample buffer with 10 mM DTT) in boiling water for 5 min and beads were removed from the protein elutions with a magnetic stand.

**Immunoblots**. Protein extracts were mixed with the reducing sample buffer and heated in boiling water for 5 min. Protein extracts and protein elutions from immunoprecipitations were separated by SDS-PAGE or by native 6% PAGE and transferred to a nitrocellulose membrane. Membranes were then saturated with 5% fat-free milk powder in Tris-buffered saline with 0.2% Tween-20 (TBS-T) for 30 min and incubated overnight at 4 °C with a primary antibody. Membranes were washed thrice with TBS-T and incubated for 2 h at room temperature with a secondary antibody coupled to horse radish peroxidase (Dako). After five washes of the membranes with TBS-T, protein bands were visualized with ECL (Enhanced ChemiLuminescence, Advansta). Uncropped scans are available for all the blots in the Source Data file.

**Luciferase assays**. Cells were seeded in white medium and transfected with PEI with an expression vector for ERα, GR, or PR for the experiments assessing activities of these pathways, a luciferase reporter plasmid and pRL-CMV. After 18 h, the medium was changed to fresh white medium and cells were treated for 24 h with hormones or inhibitors where indicated. Cells were then lysed using the Passive Lysis Buffer (Promega) and firefly luciferase and Renilla activities were measured in cell lysates with the Dual-Luciferase kit (Promega) with a bioluminescence plate reader. Renilla activity from pRL-CMV was used as a transfection control.

**Transferrin uptake assays**. Transfected cells were incubated in serum-free medium for 2 h to remove any remaining transferrin. Cells were then exposed to 50 μg/ml transferrin conjugated to AlexaFluor 680 (ThermoFisher) at 37 °C for 30 min. Cells were then washed three times with ice-cold PBS and fixed with 4% paraformaldehyde (PFA) for 10 min. After washing the cells at least three times with PBS, transferrin bound on the outside of cells was removed by two 30 min acid washes with 20 mM MES pH 4, 150 mM NaCl, with agitation at 4 °C. Following three washes with PBS, the coverslips were mounted on slides and imaged with a Leica LSM700 confocal microscope.

**Immunofluorescence**. To observe early endosome and lysosome trafficking, transferrin uptake assays were combined with immunostaining of EEA1 and LAMP1 markers, respectively. Briefly, following the 2 h starvation, cells were exposed to transferrin for 1 h to allow for transferrin uptake and trafficking to early endosome and lysosome. After washing the cells with PBS and PFA fixation for 10 min, cells were permeabilized with PBS containing 1% Triton X-100, 1% bovine serum albumin (BSA) for 10 min. Three percent BSA was used to block the nonspecific binding. Fifty microliters of primary antibody against EEA1 at a dilution of 1:50 and against LAMP1 diluted 1:700 with PBS containing 1% BSA were incubated with the cells for 2 h at room temperature (RT). After washing off the residual primary antibodies with PBS three times, cells were incubated with a FITC-conjugated secondary antibody (1:10,000), washed three times with PBS and mounted on slides. A Leica LSM700 confocal microscope was used for image acquisition.

**Kaplan–Meier analyses**. Kaplan–Meier curves were generated using Kaplan–Meier Plotter (kmplot.com)[87] with Vps11 or Vps18 on breast cancer or gastric cancer datasets.

**Statistical analyses**. The bar graphs show averages of several independent experiments (see legends for details) and the errors of the means; for luciferase experiments, each experiment comprised triplicate samples. Statistical significance was determined with unpaired and two-sided Student's *t*-tests.

**Reporting summary**. Further information on research design is available in the Nature Research Reporting Summary linked to this article.

## Data availability

The proteomics data are available as indicated in Methods in the ProteomeXchange Consortium via the PRIDE partner repository with identifier PXD009178. Source data for all blots and bar graphs are provided as a Source Data file for Figs. 1a, 2a, 2c–o, 3a–e, 4a, 4b, 4e–l, 5b–i and 6a–h, and Supplementary Figs. 1a–h, 2a–c, 3b–f, 4a–f, 5b, 5d–g, 6a–c, 6e. A reporting summary for this Article is available as a Supplementary Information file. The other data that support the findings of this study are available from the corresponding author upon reasonable request.

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

## Acknowledgements

We are grateful to Pierre Frey and Kamilla Malinowska for their early efforts on this project, to Manfredo Quadroni of the Protein Analysis Facility of the University of Lausanne, Switzerland, for the proteomic analyses and generous assistance with data processing, to Jean Gruenberg for his comments on the manuscript and for providing reagents, to Christoph Bauer and Jérôme Bosset for their technical help with confocal microscopy, and to numerous other colleagues mentioned in Methods for their gifts of reagents. This work was supported by the Canton de Genève, the Swiss National Science Foundation, and the Fondation Medic.

## Author contributions

G.S., R.K.M., F.K., and D.P. designed the experiments. G.S. performed the vast majority of the experiments. M.A.B. and N.M.G. performed several co-IP and membrane trafficking experiments, respectively, R.K.M. and F.K. supervised the experiments with flies, D.P.P. made the initial discovery of Vps11 in a yeast screen, P.C.E. contributed to the generation of the ERα-Vps11-Vps18 interactome. G.S., M.A.B., N.M.G., R.K.M., D.P.P., P.C.E., F.K., and D.P. analyzed the data. G.S. and D.P. wrote the paper.

## Additional information

**Competing interests:** The authors declare no competing interests.

