## [Peer Review File · Nature Communications]

Reviewers' comments:

Reviewer #1 (Remarks to the Author):

This paper explores the role of the Vps11 and Vps18 proteins in signal transduction and link them as regulators of cytoplasmic ER complex. The paper takes a very detailed and comprehensive approach to define and characterize the link between the Vps proteins and various putative targets. The fly knockouts are explored and the phenotypic changes suggest that the core Vps components are required for wing development. ER is shown to be repressed by Vps11/18 and this is shown to involve a complex with PELP1 and Src. A key residue on ER is implicated in this repression.

This is a solid paper and the results are important and will be of interest to the community. The experiments are comprehensive and the data are robust. However, there are several things that need to be addressed.

- Although the individual experiments are carefully conducted and the conclusions are therefore valid, it's unclear why all of the ER work is limited to one endogenous cell line and a non-relevant model that has ER added exogenously.
- Can the authors validate some of the key findings in another model that is ER dependent?
- Some experiments, such as the CARM1 experiment are difficult to reconcile given that they were conducted in a cell line that isn't dependent on ER. Therefore it's not clear if CARM1 and SRC1 are required or function in this model.
- Is it possible that ER co-repressor proteins (instead of the co-activators) are involved?
- David Rimm has shown that there is very little cytoplasmic ER in clinical samples (Welsh, Clin Can Res, 2012) and that any signal is from a non-specific antibody. If this is correct, is it possible that the mechanisms described here are limited to model systems? This should be discussed or acknowledged by the authors in the discussion.
- The fly work, whilst generating an interesting phenotype, feels completely unrelated to the rest of the paper and I question whether it adds anything to the paper.
- The authors conclude that 'only the phosphorylation of ER on Y537 is essential'. This needs to be reworded, because the authors only tested a few residues. It is the only one of the three tested but the text currently suggests that it is the only residue on the entire protein, which is not accurate.
- The addition of the gastric Kaplan Meier curve is redundant and doesn't add anything to the paper.

Reviewer #2 (Remarks to the Author):

The manuscript by Segala et al. characterizes the role of E3 ubiquitin ligase activities of Vps11 and Vps18, two core components of CORVET/HOPS complexes, in the regulation of diverse signaling pathways. The authors present solid evidence of the E3 enzymatic activity of Vps11 and Vps18, of which only the latter one was previously identified as a ubiquitin ligase but remained poorly characterized. They further employ an experimental strategy based on the overexpression of Vps11/Vps18 and SILAC mass spectrometry to identify potential substrates of the two enzymes. Among several pathways affected by overexpression of Vps11/Vps18, the authors focus on signaling by estrogen receptor α (ER α). They uncover a mechanism explaining a link between Vps11/18 and ER α -driven transcriptional response which is the strongest aspect of their study. Specifically, it is proposed that Vps11/18-mediated ubiquitination of the scaffold protein PELP1 represses ER α by regulating its phosphorylation by c-Src.

In general, the reported findings are novel and of broad interest to the signaling and trafficking communities of cell biologists. While the manuscript is data-rich and experimental results are mostly of high quality and convincing, some experiments or controls are required to further strengthen the conclusions from this study. Additionally, some aspects require clarification and/or justification of why certain experiments were performed. Thus, a revision is needed before the manuscript can be accepted for publication.

Major points:

1. Experiments presented in Fig. 1b require negative and positive controls. Negative – e.g. other component/s of CORVET/HOPS complexes such as Vps16 that does not harbor the RING domain; positive – any known and well characterized E3 Ub ligase. This is to compare the degree of changes in bulk ubiquitination upon Vps11/Vps18 overexpression.
2. There is no evidence that overexpression of Vps11/Vps18 does not affect endocytosis in general or trafficking of ubiquitinated cargoes toward degradation. The lack of an effect on the formation of CORVET/HOPS complexes does not exclude that increased amounts of its core components alter endosomal trafficking or fusion of early and late endosomes. Minimally, immunostaining of early and late endosomes (EEA1 and LAMP1 markers, respectively), as well as of ubiquitin upon Vps11/Vps18 overexpression should be presented.

3. Data included in Fig. 2 (*Drosophila* phenotypes upon CORVET/HOPS perturbation) are not very informative and rather disturb the flow of the argumentation. Unless better explained how this part supports the rest of the results, it can be omitted or moved entirely to the supplement.
4. There is no data on the formation of CORVET/HOPS complexes upon overexpression of Vps11/Vps18 harboring deletions or point mutations within the RING domain (Fig. 3). Moreover, similarly to point 2 above, the authors should also exclude changes in endosome maturation or degradation of ubiquitinated cargoes under these conditions.
5. The part of the manuscript describing a relationship between endocytosis and Vps11/Vps18 activity is not convincing (Fig. 4). The reported experiments lack controls in the form of functional assays showing the inhibition of endocytosis or autophagy upon knockdown of trafficking regulators or upon drug treatment (e.g. assays measuring internalization of transferrin to check the efficiency of clathrin- or dynamin-dependent endocytosis, LC3 conversion, etc.).
6. The authors should comment on the differences between Vps11 and Vps18 in their abilities to interact with ER α signaling complexes under basal or stimulated conditions (Fig. 5c). For example, it seems that in contrast to Vps18, Vps11 does not interact with ER α under basal conditions. Why is there less BCAR1 and c-Src bound to Vps18 after estradiol stimulation?
7. The data presented in Fig. 6b does not confirm the data from Fig. 5d. In input samples in Fig. 6b there is no decrease of the PELP1 level upon overexpression of Vps11/18, proposed based on the immunoblot in Fig. 5d. The authors should provide the quantification of the total PELP1 level from several experiments to draw a convincing conclusion about the effect of Vps11/18 on the PELP1 abundance. Similarly, there is a discrepancy between data in Fig. 5e and 5g (very different amounts of PELP1 co-immunoprecipitated with HA-ubiquitin upon overexpression of Vps11/18). Moreover, Fig. 5e may indicate some ubiquitination of BCAR1 by Vps11.
8. To confirm the final proposed mechanism it is necessary to check the amounts of ER α in complex with PELP1 and c-Src. The immunoblot for ER α in the immunoblot of c-Src should be added in Fig. 6b. Without this data, the conclusion “that Vps11/18 ubiquitinate PELP1 on K496 preventing c-Src from interacting with and being activated by membrane-associated ER α (Fig. 6h)” is an overstatement. Ideally, a reciprocal immunoprecipitation of ER α and blotting for PELP1/c-Src should be done in cells with endogenous versus overexpressed Vps11/18 to prove changes in the complex composition/stability.
9. The final model would be strengthened if it were verified by showing changes in the expression of natural target genes of ER α instead of only a reporter construct.
10. The part of the Discussion section referring to endosomal function (“Our work expands previous knowledge of the endosome as an essential organelle in signal transduction^{3,9,41,51}. We have provided evidence for another level of regulation at the crossroads between endosomal fusion and post-translational modification of signal transducers that further intertwines these cellular events.”) is unjustified as there are practically no data regarding endocytosis/endosomal fusion in this study.

Minor or technical points:

1. The figure legends are in many places too cryptic and difficult to understand. The number of performed experiments has to be clearly stated in each case (“several” is too enigmatic). The quantifications of band intensities in Fig. 3a or 6g probably refer only to the presented blots but instead should be based on at least three experiments.
2. The blot in Fig. 3a is not entirely convincing. In addition to the quantification of multiple experiments (see above), blotting for another core component e.g. Vps16 or unique HOPS/CORVET subunits would reinforce the conclusion on the complex formation/destabilization.
3. With regard to Fig. 3e, the conclusion: “The combination of Vps11 and Vps18 overexpression repressed ER α similarly showing that Vps11/18 affect ER α independently of each other” is not clear.
4. The abbreviation of E2 for estradiol is nowhere explained and can be confused throughout the text with E2 denoting a ubiquitin-conjugating enzyme.
5. The Methods section lacks the information about:
 - Catalog numbers and dilutions of antibodies used for immunoblotting, amounts of antibodies used for IP (information important in case of reproducing the work);
 - Concentration of compounds used for various treatments: E2, rapamycin, brefeldin A, chloroquine, MG132 (written inconsistently MG132 or MG-132), etc.; amounts of plasmid DNA used for transfection.
6. Schemes in Fig. 1a and 3b should include the number of amino acids of the presented proteins.
7. Legend to Fig. 5g has a wrong reference to panel f (should be panel e).

Reviewer #3 (Remarks to the Author):

Segala et al. provide evidence that the Vps18 and Vps11, which are members of tethering complexes, HOPS and CORVET, in endosomal fusion act also as E3 ligases. The E3 ligase activity appears to be independent of their function in membrane tethering. Moreover, they show that VPS11/18 mediated ubiquitylation impair signal transduction through ER α and c-Src.

This manuscript is in principle a nice piece of work. The problem with this study however is that the authors use rather indirect measures to conclude that Vps11 and Vps18 are E3 ligases, and that it is this activity that impinges on signaling pathways.

- Not all RING domain containing proteins have E3 ligase activity. The changes the authors see in the Silac after overexpression of Vps11 or Vps18 may not be a direct consequence of Vps11 or Vps18 being a E3 ligase. In the endosomal system, numerous E3 ligases act, and the Vps11 and Vps18 could stimulate the activity of one or more E3 ligases and also perhaps inhibit the activity of others. This would also explain why the authors observe almost as many sites less ubiquitylated than with increased ubiquitylation.
- Other components of the HOPS (Vps41) and CORVET (Vps8) complexes also contain a RING domain. Have the authors tested Vps41 as well? In particular, Vps18 appears to recruit Vps41 via the RING domain into the HOPS complex (Hunter et al., *Biochem. J.* 2017).
- A key experiment, which is missing in this analysis is a transplanted experiment. Can the RING domain of Vps11 and Vps18 replace the RING domain of a known E3 ligase? Or alternatively, can recombinant Vps11 and Vps18 ubiquitylate substrates such as PELP1 in vitro?
- HOPS and CORVET are not very stable complexes and they can interchange subunits (Peplowska et al *Dev. Cell* 2007). Thus, overexpression of mutants in the RING domain or lacking the ring domain, will also affect HOPS and CORVET function. This might be different than eliminating other components. If Vps18 for example cannot recruit Vps41, the complex can still bind Rab7 membranes one side, but the tethering function should no longer work. In addition, Vps16 and Vps33A have paralogs, VIPAS39 and Vps33B. In different systems interaction in a various combination have been reported. The authors may want to look into Vps8 and Vps41 since they have RING domains too. Vps8 is very unlikely to have E3 ligase activity because of an insertion in the RING domain.

Reviewer #4 (Remarks to the Author):

This manuscript from Segala et al describes a potential role of the Vps11/18 E3 ligases in the regulation of the estrogen receptor signaling. In this study, the authors establish that the Vps11 and Vps18 (members of the CORVET complex) have each an ubiquitin E3 ligase activity. The first employed a proteomics approach to unveil potential targets of the E3s to then use a genetic

approach in drosophila that suggests an involvement of the two E3s in the ecdysone signaling. The study then focuses on further characterizing a potential role of Vsp11/18 in regulating the estrogen receptor by modifying the PELP1 scaffold protein. I was tasked by the editor to more specifically evaluate the proteomics portion of the study.

The authors combined a SILAC approach with GlyGly enrichment of ubiquitinated peptides, a method now well established but that remains technically challenging. The approach also relies on overexpression of each of the E3 ligases, which is adequate but not without a risk of introducing an artifact. The experiments are well controlled and presented: two E3 ligases individually assessed and each experiment performed in triplicate. The methods are also generally well and sufficiently described.

Two elements should be clarified. Based on the methods, the authors indicated that they measured the total ratios of unmodified peptides. It should be more clearly stated how these results were collected (e.g. analysis of total cell lysate prior to GlyGly enrichment) and whether these values were used to normalize the H/L ratios. As well, it should be made clearer that C-term GlyGly were filtered out (if not they should, as these tend to be false identifications).

The presented proteomics analysis indicates which pathways may be impacted by the two studied E3 ligases, including signal transduction and protein degradation. Although these analyses were well executed and presented, it is not clear to me how the obtained results are linked to the rest of the manuscript. Therefore, the authors should better explain how these results are relevant in the context of their study in the discussion.

The reviewer would like to add the following two points that should not impact the presented work. Heavy labeled R is not required when analyzing GlyGly peptides (that should contain at least one K). Standard practice in proteomics journals is that for modified peptides the following are also provided: peptide sequence, search and probability scores, observed mass and mass error, charge, probability of site localization, individual SILAC ratios. However, Nature Communications has not a clear guideline; as such the authors would not be required to provide this information. The manuscript indicates that the data was deposited into PRIDE database; although it was not accessible at the time of the review.

Response to reviewers' comments regarding NCOMMS-18-14597-T

We ask you to address the reviewer comments in full and we would particularly encourage you to follow the reviewers' suggestions to further validate your results with control experiments, to present complementary evidence that confirms the central conclusions of your work, and to report the SILAC data in more detail. Furthermore, we feel that the manuscript would benefit from a clearer explanation of the relevance of the Drosophila experiments as well as the proteomics and pathway analyses, which should be set in clear context with the rest of the study.

We appreciate the wealth of highly constructive comments and are happy to provide a revised version of our manuscript. It incorporates complementary evidence as new data and figures, all of which supports and confirms our conclusions, and more information on the SILAC data (and proper access information for reviewers, below in response to reviewer #4). In our point-by-point responses below, we give a better explanation of why we would like to keep the Drosophila data in the paper and in the Results section, we have edited the text to smoothen that particular transition.

Reviewer #1:

- *Although the individual experiments are carefully conducted and the conclusions are therefore valid, it's unclear why all of the ER work is limited to one endogenous cell line and a non-relevant model that has ER added exogenously.*
- *Can the authors validate some of the key findings in another model that is ER dependent?*

We have used MCF7 cells to validate the key findings that we had obtained with two unrelated cell lines. These new data are included as Fig. 3i and Fig. 6c.

- *Some experiments, such as the CARM1 experiment are difficult to reconcile given that they were conducted in a cell line that isn't dependent on ER. Therefore it's not clear if CARM1 and SRC1 are required or function in this model.*

293T cells have been extensively used in the nuclear receptor / ER α field to characterize the nuts and bolts of nuclear receptor / ER α functions at the cellular and molecular levels. For example, we had used HEK293T cells to characterize the involvement of CARM1 in ligand-independent activation of ER α (Carascossa et al., Genes Dev, 2010).

- *Is it possible that ER co-repressor proteins (instead of the co-activators) are involved?*

Formally yes, but we have now added a result that shows that Vps11/18 do not work through the ER α co-repressor NCoR. Although NCoR does repress ER α activity, Vps11/18 overexpression still inhibits. These new data are included as Supplementary Fig. 4g.

- *David Rimm has shown that there is very little cytoplasmic ER in clinical samples (Welsh, Clin Can Res, 2012) and that any signal is from a non-specific antibody. If this is correct, is it possible that the mechanisms described here are limited to model*

systems? This should be discussed or acknowledged by the authors in the discussion.

There may indeed be only little, but the little there is appears to be functionally relevant. There is ample evidence from pharmacological and KO and KI experiments in the mouse that non-genomic signaling by ER α accounts for some ER α functions. We have added a second authoritative review on this subject in our Introduction: Arnal et al. (2017) *Physiol. Rev.* 97, 1045.

- The fly work, whilst generating an interesting phenotype, feels completely unrelated to the rest of the paper and I question whether it adds anything to the paper.

We understand that there is no obvious direct link between the fly work and our characterization of the effects on ER α . However, our story starts with the characterization of the E3 ligase activities of Vps11/18 and their impact and that of their HOPS/CORVET complexes on signal transduction. The data of Figures 1-3 are there to demonstrate the tight link between these components and several signaling pathways both in flies and in mammalian cell lines. At this point in the story, ER α is just one of several signaling molecules that is affected. And indeed, different signaling pathways are affected in different ways. To clarify the transition, we have now expanded our statement in the Results section, before referring specifically to ER α results. We would like to argue that the first part of the paper is on the general connection between Vps11/Vps18 and signaling and the second part dissects one signaling pathway in particular in much more detail.

- The authors conclude that 'only the phosphorylation of ER on Y537 is essential'. This needs to be reworded, because the authors only tested a few residues. It is the only one of the three tested but the text currently suggests that it is the only residue on the entire protein, which is not accurate.

We have now reworded it, but we would like to point out that we tested 6 sites altogether.

- The addition of the gastric Kaplan Meier curve is redundant and doesn't add anything to the paper.

We respectfully disagree. The data for gastric cancer, not known to be regulated by ER α , constitutes a highly interesting outgroup since it displays the inverse correlation with Vps11/18 expression.

Reviewer #2:

Major points:

1. Experiments presented in Fig. 1b require negative and positive controls. Negative – e.g. other component/s of CORVET/HOPS complexes such as Vps16 that does not harbor the RING domain; positive – any known and well characterized E3 Ub ligase. This is to compare the degree of changes in bulk ubiquitination upon Vps11/Vps18 overexpression.

We have repeated the experiment of Fig. 1b (new panel) with Vps16 and Vps33A as additional components. Moreover, there are plenty of negative controls in Fig. 3c. In contrast, since we do show the existence of specific Vps11/18 ubiquitinomes, we do not see how an unrelated E3 ligase could be a relevant control.

2. There is no evidence that overexpression of Vps11/Vps18 does not affect endocytosis in general or trafficking of ubiquitinated cargoes toward degradation. The lack of an effect on the formation of CORVET/HOPS complexes does not exclude that increased amounts of its core components alter endosomal trafficking or fusion of early and late endosomes. Minimally, immunostaining of early and late endosomes (EEA1 and LAMP1 markers, respectively), as well as of ubiquitin upon Vps11/Vps18 overexpression should be presented.

We have performed all of these control experiments and these data are now shown in Fig. 4c and d. They show that Vps11/18 overexpression does not affect endocytosis nor membrane traffic from early endosomes to lysosomes.

3. Data included in Fig. 2 (Drosophila phenotypes upon CORVET/HOPS perturbation) are not very informative and rather disturb the flow of the argumentation. Unless better explained how this part supports the rest of the results, it can be omitted or moved entirely to the supplement.

We paste here again our response to a similar comment of reviewer #1: We understand that there is no obvious direct link between the fly work and our characterization of the effects on ER α . However, our story starts with the characterization of the E3 ligase activities of Vps11/18 and their impact and that of their HOPS/CORVET complexes on signal transduction. The data of Figures 1-3 are there to demonstrate the tight link between these components and several signaling pathways both in flies and in mammalian cell lines. At this point in the story, ER α is just one of several signaling molecules that is affected. And indeed, different signaling pathways are affected in different ways. To clarify the transition, we have now expanded our statement in the Results section, before referring specifically to ER α results. We would like to argue that the first part of the paper is on the general connection between Vps11/Vps18 and signaling and the second part dissects one signaling pathway in particular in much more detail.

4. There is no data on the formation of CORVET/HOPS complexes upon overexpression of Vps11/Vps18 harboring deletions or point mutations within the RING domain (Fig. 3). Moreover, similarly to point 2 above, the authors should also exclude changes in endosome maturation or degradation of ubiquitinated cargoes under these conditions.

Now there are data regarding both of these issues. Our new Supplementary Fig. 3c and Supplementary Fig. 4c show that there are no effects.

5. The part of the manuscript describing a relationship between endocytosis and Vps11/Vps18 activity is not convincing (Fig. 4). The reported experiments lack controls in the form of functional assays showing the inhibition of endocytosis or autophagy upon knockdown of trafficking regulators or upon drug treatment (e.g.

assays measuring internalization of transferrin to check the efficiency of clathrin- or dynamin-dependent endocytosis, LC3 conversion, etc.).

We have added these control experiments as panels a and f in Supplementary Fig. 4 for endocytosis and autophagy, respectively. They show that our dynamin mutant is indeed a mutant and blocks endocytosis, and that autophagy is induced by our treatments but not affected by Vps11/18 overexpression.

6. The authors should comment on the differences between Vps11 and Vps18 in their abilities to interact with ER α signaling complexes under basal or stimulated conditions (Fig. 5c). For example, it seems that in contrast to Vps18, Vps11 does not interact with ER α under basal conditions. Why is there less BCAR1 and c-Src bound to Vps18 after estradiol stimulation?

There is obviously some biological variation, which we cannot explain in detail. We cannot exclude that there are some differences between Vps11 and Vps18, which have yet to be characterized (beyond the scope of this paper).

7. The data presented in Fig. 6b does not confirm the data from Fig. 5d. In input samples in Fig. 6b there is no decrease of the PELP1 level upon overexpression of Vps11/18, proposed based on the immunoblot in Fig. 5d. The authors should provide the quantification of the total PELP1 level from several experiments to draw a convincing conclusion about the effect of Vps11/18 on the PELP1 abundance. Similarly, there is a discrepancy between data in Fig. 5e and 5g (very different amounts of PELP1 co-immunoprecipitated with HA-ubiquitin upon overexpression of Vps11/18). Moreover, Fig. 5e may indicate some ubiquitination of BCAR1 by Vps11.

The experiments of Fig. 5d and 6b are totally different and have a different timing. We have added a statement in the legends to Fig. 5d and Fig. 6b to clarify this. Fig. 5e and 5g are also different types of experiments, the former being with endogenous PELP1 and the latter with transfected wild-type PELP1 serving as a control for the mutant K496R. As for BCAR1, we have not seen any clear bands indicating the presence of a ubiquitinated form.

8. To confirm the final proposed mechanism it is necessary to check the amounts of ER α in complex with PELP1 and c-Src. The immunoblot for ER α in the immunoblot of c-Src should be added in Fig. 6b. Without this data, the conclusion “that Vps11/18 ubiquitinate PELP1 on K496 preventing c-Src from interacting with and being activated by membrane-associated ER α (Fig. 6h)” is an overstatement. Ideally, a reciprocal immunoprecipitation of ER α and blotting for PELP1/c-Src should be done in cells with endogenous versus overexpressed Vps11/18 to prove changes in the complex composition/stability.

Indeed, it was important to demonstrate that the core complex of ER α with c-Src is disrupted upon overexpression of Vps11/18. This being a pivotal point, we have added these data both for MDA-MB-134 and MCF7 cells as a new Fig. 6c.

9. The final model would be strengthened if it were verified by showing changes in the expression of natural target genes of ER α instead of only a reporter construct.

We would like to point out that we did show in Fig. 3g how the knock-down of Vps11/18 affects endogenous ER α target genes. We have now added an experiment where we overexpressed Vps11/18 both in MDA-MB-134 and MCF-7 cells to look at the impact on endogenous ER α target genes (new Fig. 3h-i). Moreover, we now show that PELP1 K496R impairs the repression of endogenous ER α target genes by the overexpression of Vps11/18 (new Fig. 5i).

10. The part of the Discussion section referring to endosomal function (“Our work expands previous knowledge of the endosome as an essential organelle in signal transduction^{3,9,41,51}. We have provided evidence for another level of regulation at the crossroads between endosomal fusion and post-translational modification of signal transducers that further intertwines these cellular events.”) is unjustified as there are practically no data regarding endocytosis/endosomal fusion in this study.

Although we have now added a whole series of control experiments regarding endocytosis and endosomal fusion, we have added a section in the Discussion about these issues and to speculate that Vps11/18 may be moonlighting as E3 ligases independently of their functions as components of HOPS/CORVET.

Minor or technical points:

1. The figure legends are in many places too cryptic and difficult to understand. The number of performed experiments has to be clearly stated in each case (“several” is too enigmatic). The quantifications of band intensities in Fig. 3a or 6g probably refer only to the presented blots but instead should be based on at least three experiments.

We have edited the legends and added additional details wherever reasonable.

2. The blot in Fig. 3a is not entirely convincing. In addition to the quantification of multiple experiments (see above), blotting for another core component e.g. Vps16 or unique HOPS/CORVET subunits would reinforce the conclusion on the complex formation/destabilization.

These are native gels and rather challenging to do. This particular one is representative of several. Please note that we did not blot for Vps11/18, but for another Vps-C component (Vps33A). Doing it for Vps16 would be totally redundant and would not provide any new insights. Moreover, as mentioned above, we now show an additional one in Supplementary Fig. 3c.

3. With regard to Fig. 3e, the conclusion: “The combination of Vps11 and Vps18 overexpression repressed ER α similarly showing that Vps11/18 affect ER α independently of each other” is not clear.

We have reworded this sentence.

4. The abbreviation of E2 for estradiol is nowhere explained and can be confused throughout the text with E2 denoting a ubiquitin-conjugating enzyme.

Thank you for spotting this. We now properly introduce the abbreviation E2 and we have avoided the use of the term E2 for ubiquitin/sumo-conjugating enzymes.

5. The Methods section lacks the information about:

- Catalog numbers and dilutions of antibodies used for immunoblotting, amounts of antibodies used for IP (information important in case of reproducing the work);
- Concentration of compounds used for various treatments: E2, rapamycin, brefeldin A, chloroquine, MG132 (written inconsistently MG132 or MG-132), etc.; amounts of plasmid DNA used for transfection.

Most of the relevant information is (and was) in the Methods section as well as the Reporting Summary required by the publisher. This did indeed not include the drug concentrations, which we have now added.

As for the drug name MG132, it appears exactly twice: once in the text and once in Supplementary Fig 4 and in both instances, it is written exactly the same way.

6. Schemes in Fig. 1a and 3b should include the number of amino acids of the presented proteins.

These additional details would only clutter the figures and not provide any relevant information.

7. Legend to Fig. 5g has a wrong reference to panel f (should be panel e).

Thanks, this has now been fixed.

Reviewer #3:

- Not all RING domain containing proteins have E3 ligase activity. The changes the authors see in the Silac after overexpression of Vps11 or Vps18 may not be a direct consequence of Vps11 or Vps18 being a E3 ligase. In the endosomal system, numerous E3 ligases act, and the Vps11 and Vps18 could stimulate the activity of one or more E3 ligases and also perhaps inhibit the activity of others. This would also explain why the authors observe almost as many sites less ubiquitylated than with increased ubiquitylation.

We have added a comment to that effect at the end of the first paragraph of the Results.

- Other components of the HOPS (Vps41) and CORVET (Vps8) complexes also contain a RING domain. Have the authors tested Vps41 as well? In particular, Vps18 appears to recruit Vps41 via the RING domain into the HOPS complex (Hunter et al., Biochem. J. 2017).

Thank you for this suggestion. We have now done this experiment with both Vps8 and Vps41 (new Supplementary Fig. 3I) and found that they do not affect ER α activity.

- A key experiment, which is missing in this analysis is a transplantation experiment. Can the RING domain of Vps11 and Vps18 replace the RING domain of a known E3

ligase? Or alternatively, can recombinant Vps11 and Vps18 ubiquitylate substrates such as PELP1 in vitro?

We are at a loss to understand what could be gained by a transplantation experiment. As for demonstrating ubiquitination of PELP1 in vitro with purified components, this is really beyond the scope of this study. We maintain that the weight of the collective (in vivo) evidence is strong enough to conclude that PELP1 is directly ubiquitinated by Vps11/18 in a K496-dependent fashion.

- HOPS and CORVET are not very stable complexes and they can interchange subunits (Peplowska et al Dev. Cell 2007). Thus, overexpression of mutants in the RING domain or lacking the ring domain, will also affect HOPS and CORVET function. This might be different than eliminating other components. If Vps18 for example cannot recruit Vps41, the complex can still bind Rab7 membranes one side, but the tethering function should no longer work. In addition, Vps16 and Vps33A have paralogs, VIPAS39 and Vps33B. In different systems interaction in a various combination have been reported. The authors may want to look into Vps8 and Vps41 since they have RING domains too. Vps8 is very unlikely to have E3 ligase activity because of an insertion in the RING domain.

We would like to point out that with the revised manuscript we clearly demonstrate that the Vps-C complex is not perturbed by Vps11/18 RING mutants and that membrane traffic isn't either. Furthermore, we did the suggested experiment with Vps8 (see above) despite the fact that the reviewer seems to say here that it is unlikely to be worth it.

Reviewer #4:

Two elements should be clarified. Based on the methods, the authors indicated that they measured the total ratios of unmodified peptides. It should be more clearly stated how these results were collected (e.g. analysis of total cell lysate prior to GlyGLy enrichment) and whether these values were used to normalize the H/L ratios.

We have now added a reference in Methods to a publication by Quadroni and colleagues, which describes the method in more detail. The procedure reproduced in our Methods is largely similar to the one by Quadroni and colleagues. Total cell lysates were not analysed separately to determine total protein ratios. Note that GlyGly peptide enrichments are always partial and the samples obtained always contain a considerable number of unmodified peptides. So the MS analysis produces (as a by-product) quantitation of a number of proteins, with an obvious bias towards abundant ones. Overall, MS and MaxQuant analysis produced 19366 peptide identifications (including differently modified forms of the same sequence). Of these, 13781 were unmodified peptides while 5137 were peptides with 1, 2 or 3 GlyGly modification(s). The total protein ratios acquired internally in each sample were then used by the MaxQuant software for global normalization of GlyGly site SILAC ratios. It should be noted that this is a normalization that corrects for bias in loading amounts, not for biological variation in quantity of individual proteins.

As well, it should be made clearer that C-term GlyGly were filtered out (if not they should, as these tend to be false identifications).

We assume that the reviewer is referring to the situation in which a GlyGly modification is identified on the C-terminal Lys residue of a peptide (if he/she is referring to the C-termini of proteins then the comment is not clear to us). We checked and in our list of GlyGly sites we found only 15 out of 4868 sites (0.3%) containing a single Lys with the structure (-----K*), where K* is a GlyGly-modified Lys. We therefore conclude that this is not a major issue in our dataset. It should also be considered that it is difficult to exclude categorically that trypsin cannot cleave at such sites.

The presented proteomics analysis indicates which pathways may be impacted by the two studied E3 ligases, including signal transduction and protein degradation. Although these analyses were well executed and presented, it is not clear to me how the obtained results are linked to the rest of the manuscript. Therefore, the authors should better explain how these results are relevant in the context of their study in the discussion.

The characterization of the Vps11/18 ubiquitinomes led us to consider the regulation of "signal transduction" as a relevant function. We state this explicitly in the Abstract and in the Results section before we move on to examine a panel of signaling pathways in flies and mammalian cells. Hence, the proteomics is highly relevant in that this is what led us to demonstrate that Vps11/18 do regulate a variety of signaling pathways.

The reviewer would like to add the following two points that should not impact the presented work. Heavy labeled R is not required when analyzing GlyGly peptides (that should contain at least one K). Standard practice in proteomics journals is that for modified peptides the following are also provided: peptide sequence, search and probability scores, observed mass and mass error, charge, probability of site localization, individual SIALC ratios. However, Nature Communications has not a clear guideline; as such the authors would not be required to provide this information. The manuscript indicates that the data was deposited into PRIDE database; although it was not accessible at the time of the review.

Our apologies for not having properly transmitted the reviewer access to the data in PRIDE. It is as follows: web page: <https://www.ebi.ac.uk/pride/archive/login> (note that it is important to use this web site for temporary access, NOT the standard site www.proteomexchange.org);
username: reviewer80939@ebi.ac.uk; password: DrFAI8td

Reviewers' comments:

Reviewer #1 (Remarks to the Author):

The authors have addressed my concerns with new data, new analyses or a logical argument. As such, I am happy with the revised version.

Reviewer #2 (Remarks to the Author):

The revised manuscript by Segala et al has satisfactorily addressed most of my concerns. Still, I remain unconvinced about three of my previously raised issues:

- major point 3: despite some rewording of the text, the relevance of *Drosophila* work to the rest of the story is at best questionable (the authors have decided to keep the results as one of the main figures). The fly phenotypes raise more questions, rather than clarify any of the main findings of the study.
- major point 7, Figs. 5D and 6B: the effect of Vps11/18 overexpression on the total protein level of PELP1 remains unclear. The authors have not provided a quantification of PELP1 levels from several experiments, as requested. Instead, they speculate without providing any proof that lower level of PELP1 visualized in Fig. 5D reflects increased degradation due to ubiquitination of PELP1 upon 6 days of Vps11/18 overexpression. They further argue that the unchanged level of PELP1 in Fig. 6B results from the too short time (3 days) of Vps11/18 overexpression but nevertheless they observe the destabilization of Src-ER complex already at this time point. If the impact of Vps11/18 overexpression on the PELP1 abundance depends on time, e.g. due to the long lifetime of PELP1 protein before its degradation, then the authors should properly document it in a quantitative manner. This is one of the key results to support their final model and thus should be convincingly proven.
- minor point 1, regarding data reporting standards: the authors should state clearly the number of biological repeats of performed experiments. In most figure legends, a statement about at least two biological repeats appears, which is not appropriate (in Materials and Methods the authors still write about "several independent experiments with technical replicates"). Moreover, the authors did not check the distribution of data prior to statistical testing to select an appropriate test accordingly (or there is no information about such procedure in Materials and Methods).

Reviewer #3 (Remarks to the Author):

The authors have significantly improved the manuscript and addressed the concerns satisfactorily.

Reviewer #4 (Remarks to the Author):

This is a resubmission of a manuscript from Segala et al dealing with the characterization of the Vsp11/Vsp18 subunits of the HOPS-CORVET complex. I was initially asked to mostly review the proteomic component of the study; therefore, I will refrain to comment on other sections as three other reviewers have done so.

The authors should modify the following elements Figure 1 that I missed in my first review. In figure 1f, proteins should not be listed as “substrates” but instead as “proteins affected by...” Proteins with lower ubiquitination levels upon overexpression of an E3 cannot be considered as substrates. The scale in Figure 1h should be changed. Blue does not represent proteins enriched in “control” but proteins for which ubiquitination was reduced. The legend should also explain what influenced the size of the nodes.

I am still confused with the structure of the manuscript. **two** other reviewers raised the same concern and, in my opinion, this was not properly addressed in the revised version. The presented proteomic analysis remains disconnected to the rest of the study. Signal transduction is a very vague GO term, as there are numerous possible pathways, especially since this was one of many terms identified (e.g., protein degradation, metabolism...). The rationale for testing the fly phenotypes is seemingly driven by literature (ref 38-39) and not the proteomic work. Moreover, the wing morphological defects could be driven by many different events. More importantly, it is not specific to Vsp11/18 but also to Vps16 and 33 that do not have any RING domain. If there is no clear and direct link between the targets identified by mass spectrometry and the observed phenotypes; the authors could instead use these results to establish the physiological importance of VPS-c in Figure 1, and then proceed to show that Vps11/18 have an apparent E3-ligase activity (regroup Fig 1a-b with Fig 3a-c) before moving to the proteomic analysis. Following the proteomic study, the authors should more clearly explain why reporters for ERalpha, NFKappaB, TCF, etc... were specifically selected (in Fig 3). Are they meant to randomly represent different pathways regulated in

the cell? Or, were there specific proteins targeted by these pathways identified by mass spectrometry? It would be particularly important to better link the proteomic analysis to ERalpha.

Response to reviewers' comments regarding NCOMMS-18-14597A

To address the general impression of disjointedness between the different parts of our story, we have revised the text in multiple places to try to explain and to smoothen the transitions. The fly work has been moved to the Supplementary. These revisions led us to reshuffle some of the figures, but the data remain unchanged.

Reviewer #2:

The revised manuscript by Segala et al has satisfactorily addressed most of my concerns. Still, I remain unconvinced about three of my previously raised issues: - major point 3: despite some rewording of the text, the relevance of Drosophila work to the rest of the story is at best questionable (the authors have decided to keep the results as one of the main figures). The fly phenotypes raise more questions, rather than clarify any of the main findings of the study.

You definitely had a point, and hence, we have now moved the Drosophila work to a Supplementary Figure. Accordingly, its place in the text has changed and its importance in the text (including the Abstract) has been reduced. We are convinced that it still provides an interesting complementary piece of information. Unlike the mammalian signaling pathways we go on to study in detail, wing development in the fly requires most or all components of the HOPS/CORVET complexes and thus most likely their standard functions in membrane traffic. Although more work will have to be done to dissect this mechanistically, this illustrates that HOPS/CORVET complexes and components can modulate signaling in multiple different ways.

- major point 7, Figs. 5D and 6B: the effect of Vps11/18 overexpression on the total protein level of PELP1 remains unclear. The authors have not provided a quantification of PELP1 levels from several experiments, as requested. Instead, they speculate without providing any proof that lower level of PELP1 visualized in Fig. 5D reflects increased degradation due to ubiquitination of PELP1 upon 6 days of Vps11/18 overexpression. They further argue that the unchanged level of PELP1 in Fig. 6B results from the too short time (3 days) of Vps11/18 overexpression but nevertheless they observe the destabilization of Src-ER complex already at this time point. If the impact of Vps11/18 overexpression on the PELP1 abundance depends on time, e.g. due to the long lifetime of PELP1 protein before its degradation, then the authors should properly document it in a quantitative manner. This is one of the key results to support their final model and thus should be convincingly proven.

PELP1 expression levels or turnover rates by themselves are not relevant since overexpression does not suppress the effects and disruption of PELP1 complexes precedes PELP1 turnover. While we had hoped that this would already be apparent in version R1 of our manuscript, we have added another sentence to make it clearer. In the chapter "Vps11/18 repress ER α " (page 13), we added the sentence that starts with "Importantly, this result also points out that variations of the PELP1 levels, at least within certain limits, do not account for the effects....", complementing the sentence a few lines down that states "... in these short-term experiments even before it leads to its degradation...".

In response to the Editor's additional comments regarding this issue, we have now addressed it by performing a time-course experiment. The new data are shown in Supplementary Fig. 6a and the text has been adapted accordingly. The experiment confirms that degradation of PELP1 takes a long time and thus, that the key event is the short-term ubiquitination of PELP1 and not the ensuing long-term degradation. We have further emphasized this by inserting a sentence in the Discussion (page 15).

- minor point 1, regarding data reporting standards: the authors should state clearly the number of biological repeats of performed experiments. In most figure legends, a statement about at least two biological repeats appears, which is not appropriate (in Materials and Methods the authors still write about "several independent experiments with technical replicates"). Moreover, the authors did not check the distribution of data prior to statistical testing to select an appropriate test accordingly (or there is no information about such procedure in Materials and Methods).

We have now added these details in all legends, complementing the details given in Methods (their last chapter).

Reviewer #4:

This is a resubmission of a manuscript from Segala et al dealing with the characterization of the Vsp11/Vsp18 subunits of the HOPS-CORVET complex. I was initially asked to mostly review the proteomic component of the study; therefore, I will refrain to comment on other sections as three other reviewers have done so.

The authors should modify the following elements Figure 1 that I missed in my first review. In figure 1f, proteins should not be listed as "substrates" but instead as "proteins affected by..." Proteins with lower ubiquitination levels upon overexpression of an E3 cannot be considered as substrates. The scale in Figure 1h should be changed. Blue does not represent proteins enriched in "control" but proteins for which ubiquitination was reduced. The legend should also explain what influenced the size of the nodes.

We have removed the term substrates from Figure 1 (and worded it more carefully in the text, too). The size of the nodes is explained more clearly in the revised legend. However, the scale of Figure 1h was perfectly correct, but an explanatory comment is now added to the legend. Note that the terminology comes from GSEA.

*I am still confused with the structure of the manuscript. **two** other reviewers raised the same concern and, in my opinion, this was not properly addressed in the revised version. The presented proteomic analysis remains disconnected to the rest of the study. Signal transduction is a very vague GO term, as there are numerous possible pathways, especially since this was one of many terms identified (e.g., protein degradation, metabolism...). The rationale for testing the fly phenotypes is seemingly driven by literature (ref 38-39) and not the proteomic work. Moreover, the wing morphological defects could be driven by many different events. More importantly, it is not specific to Vsp11/18 but also to Vps16 and 33 that do not have any RING domain. If there is no clear and direct link between the targets identified by mass*

spectrometry and the observed phenotypes; the authors could instead use these results to establish the physiological importance of VPS-c in Figure 1, and then proceed to show that Vps11/18 have an apparent E3-ligase activity (regroup Fig 1a-b with Fig 3a-c) before moving to the proteomic analysis. Following the proteomic study, the authors should more clearly explain why reporters for ERalpha, NFKappaB, TCF, etc... were specifically selected (in Fig 3). Are they meant to randomly represent different pathways regulated in the cell? Or, were there specific proteins targeted by these pathways identified by mass spectrometry? It would be particularly important to better link the proteomic analysis to ERalpha.

- Choice of GO term: We are well aware that GO terms, and the annotations relating to them, can be rather vague. Nevertheless, a GO analysis is a reasonable and often used way to crunch large datasets for prioritization. In Fig. 1h, "signal transduction" is the largest cluster where all nodes are regulated the same way. Together with arguments layed out in several sentences at the end of the first Results chapter, we believe that this justifies our choice and *connects* these analyses to the next parts of the manuscript.
- Fly phenotypes: please see our responses to Reviewer #2 regarding the Drosophila work.
- The flow and structure of the manuscript have been considerably amended. Initially, we start out with a broad approach and then eventually focus on one (ER α) to go into more mechanistic details. Please see our responses to Reviewer #2 for more details.

REVIEWERS' COMMENTS:

Reviewer #2 (Remarks to the Author):

The authors have addressed my concerns. The logic of the manuscript is now also improved.